# Priority areas to protect mangroves and maximise ecosystem services

Alvise Dabalà [1,2,3] ✉, Farid Dahdouh-Guebas [2,3,4,5], Daniel C. Dunn [1,6], Jason D. Everett [1,7,8], Catherine E. Lovelock [1,4], Jeffrey O. Hanson [9], Kristine Camille V. Buenafe [1,6,7], Sandra Neubert [1,6,10] & Anthony J. Richardson [1,6,7]

Anthropogenic activities threaten global biodiversity and ecosystem services. Yet, area-based conservation efforts typically target biodiversity protection whilst minimising conflict with economic activities, failing to consider ecosystem services. Here we identify priority areas that maximise both the protection of mangrove biodiversity and their ecosystem services. We reveal that despite 13.5% of the mangrove distribution being currently strictly protected, all mangrove species are not adequately represented and many areas that provide disproportionally large ecosystem services are missed. Optimising the placement of future conservation efforts to protect 30% of global mangroves potentially safeguards an additional 16.3 billion USD of coastal property value, 6.1 million people, 1173.1 Tg C, and 50.7 million fisher days yr⁻¹. Our findings suggest that there is a pressing need for including ecosystem services in protected area design and that strategic prioritisation and coordination of mangrove conservation could provide substantial benefits to human wellbeing.

Ecosystems provide many services to humanity, including food provisioning, air and water purification and climate change mitigation[1]. These ecosystem services are threatened by global biodiversity loss caused by anthropogenic activities such as urbanisation, agriculture and fishing[2]. In 2011, a total of 194 countries agreed, via the Convention of Biological Diversity Aichi Target 11, that by 2020 they would protect at least 17% of their terrestrial systems and inland waters, and 10% of their coastal zones and marine regions, by implementing protected areas and other effective area-based conservation measures[3]. Despite failing to meet these targets, the first draft of the post-2020 Global Biodiversity Framework calls for 30% of the Earth to be protected to conserve biodiversity and enhance nature's contribution to people through ecosystem services[4].

Historically, conservation has neglected ecosystem services[5]. Although efforts have been made to calculate the value of ecosystem services provided by nature (e.g., Costanza et al.[6]; Costanza et al.[7]), data on ecosystem services are rarely included in conservation plans. Most studies focus on adequately representing biodiversity whilst minimising conflict with economic activities, such as avoiding population centres, arable land or valuable fishing areas[8,9]. However,

¹School of the Environment, The University of Queensland, St Lucia, QLD, Australia. ²Systems Ecology and Resource Management Research Unit (SERM), Department of Organism Biology, Université Libre de Bruxelles - ULB, Av. F.D. Roosevelt 50, CPi 264/1, 1050 Brussels, Belgium. ³Ecology & Biodiversity, Laboratory of Plant Biology and Nature Management, Biology Department, Vrije Universiteit Brussel - VUB, Pleinlaan 2, VUB-APNA-WE, 1050 Brussels, Belgium. ⁴Mangrove Specialist Group (MSG), Species Survival Commission (SSC), International Union for the Conservation of Nature (IUCN), Zoological Society of London, London, UK. ⁵Interfaculty Institute of Social-Ecological Transitions, Université Libre de Bruxelles - ULB, Av. F.D. Roosevelt 50, 1050 Brussels, Belgium. ⁶Centre for Biodiversity and Conservation Science (CBCS), The University of Queensland, St Lucia, QLD, Australia. ⁷Commonwealth Scientific and Industrial Research Organization (CSIRO) Environment, Queensland Biosciences Precinct (QBP), St Lucia, QLD, Australia. ⁸Centre for Marine Science and Innovation (CMSI), The University of New South Wales, Sydney, NSW, Australia. ⁹Department of Biology, Carleton University, Ottawa, Canada. ¹⁰Institute of Computer Science, Leipzig University, Leipzig, Germany. ✉e-mail: alvise.dab@gmail.com

recently there has been increasing interest in designing spatial planning approaches to maintain and enhance ecosystem services[10–13]. For example, Jung et al.[10] used spatial prioritisation to identify areas on land that protect both biodiversity and ecosystem services such as carbon sequestration and freshwater provision. Sievers et al.[13] identified areas of co-occurrence of mangrove biodiversity and ecosystem services. Similarly, Sala et al.[12] identified co-benefits in the ocean, concentrating on protecting biodiversity whilst minimising carbon remineralisation from trawling and maximising fisheries benefits, although aspects of that analysis have been criticised[14]. At local and regional scale, the same trend is observed, with multiple studies including ecosystem services in their analysis or evaluating trade-offs between ecosystem services[15–17]. Despite this increasing interest, few studies have identified priority areas in coastal systems for safeguarding biodiversity and ecosystem services.

Mangroves are at the interface between the land and the sea, providing several key ecosystem services[18,19]. The complex system of roots and branches of mangroves helps protect coastal populations and properties from flooding, strong winds and extreme weather[20,21]. Mangrove roots also reduce erosion and increase accretion, promoting stabilisation and establishment of coastal soil[20]. Additionally, mangroves store carbon over the short term in their biomass and over the long term in sediments[22]. The carbon storage of mangroves ($0.096 \, \mathrm{Tg \, C \, km^{-2}}$) is greater than that of other ecosystems on land (rainforests $0.024 \, \mathrm{Tg \, C \, km^{-2}}$; peat swamps $0.041 \, \mathrm{Tg \, C \, km^{-2}}$) and in the sea (saltmarshes $0.059 \, \mathrm{Tg \, C \, km^{-2}}$; seagrasses $0.014 \, \mathrm{Tg \, C \, km^{-2}}$)[22]. Finally, the high productivity and physical complexity of mangrove ecosystems make them important nursery grounds for many commercial and non-commercial fish and invertebrate species, providing employment and food for millions of people[23]. Although in the current analysis we focused on these ecosystem services (i.e., coastal protection, carbon storage and nursery grounds for fisheries), mangroves provide many other services, including tourism, wood and non-timber forest products, recreation, socio-cultural services and water filtration and several disservices (negative effects of nature on people), including those related to health, safety and security, leisure and recreation, and materials (see Dahdouh-Guebas et al.[18] for more details). Although mangroves clearly provide substantial benefits to people, priority areas for protection to conserve biodiversity and ecosystem services have not been identified globally.

Here we: (1) assess how well current protected areas represent mangrove biodiversity and ecosystem services; (2) identify areas that, if efficiently protected, could ensure the conservation of mangrove biodiversity and the most ecosystem services, while selecting the smallest area possible and (3) identify priority areas that could be considered for the expansion of current mangrove protection. Targeting protection in these areas of high biodiversity and ecosystem services could provide substantial social, ecological and economic benefits[24], supporting coastal communities and livelihoods. Protection of mangrove ecosystems has many positive effects, including species conservation, preservation of carbon sinks, and enhanced spillover effects leading to increased catch outside of protected areas[25,26]. Area-based conservation efforts can protect mangroves from clearing[27], maintain the ecosystem services provided, and enhance resilience to climate change[11,28]. Our analysis considers ecosystem services, including coastal protection to reduce future disaster risk[21], carbon storage to preserve carbon sinks and future carbon sequestration[29,30], and fisheries benefits to guarantee food security for local communities[31]. To evaluate the benefit of mangroves to fisheries, our approach does not consider the closure of ocean areas from fishing but just the protection of mangroves from clearing to preserve their functioning and the increase in catches that result. Importantly, the focus on ecosystem services, rather than concentrating on minimising potential economic conflicts with industry, seeks to prioritise benefits to local people. Although global conservation analyses have been criticised for poor

performance at local scales[32], and lack of consideration of impacts on local livelihoods and data justice issues[33], we believe there is utility in undertaking prioritisation at multiple scales[34], from both a community-driven and scientific perspective[35]. While the current study does not purport to direct management at local scales, it could provide several insights: guidance on how we are meeting societal goals and identifying gaps in mangrove conservation; highlighting potential advantages of mangrove conservation to many nations; informing priorities and resources deployed by non-Governmental organisations; and stimulating local and regional spatial planning with informed and prior consultation of local stakeholders including indigenous peoples[36,37].

Using the prioritizr R package[38], we generated spatial prioritisations based on mangrove ecosystem service data from the best available global datasets[21,29–31]. The prioritisations maximised the protection of 65 mangrove species (see "Methods" section). The analysis was conducted on hexagonal planning units with a spatial resolution of 20-km alongshore. Because there is no accepted definition of what constitutes protection, we have used two separate definitions for existing protection based on the IUCN categories in the World Database on Protected Areas (WDPA)[39]: strictly protected areas based on IUCN categories I-IV; and less strictly protected areas based on all IUCN categories from I to VI and unclassified areas. Our prioritisations ensured ecological representation by including fine-scale maps of mangrove cover from Global Mangrove Watch[40] at 25 m resolution, intersected with mangrove species distributions from IUCN[41]. To account for the variability in structure, height, and species diversity of mangroves across different climates, tidal amplitudes and geomorphic settings, we split the species distribution of each of the 65 mangrove species across the biophysical typologies[42] and marine provinces of the world[43], resulting in 944 sub-species distributions each assigned a specific conservation target. We assigned conservation targets (i.e., the minimum area to protect for each species) based on the area of the global distribution of mangrove species using a $\log_{10}$-linear interpolation following Rodrigues et al.[44] (see "Methods" section). Hence, species with larger distribution ranges were assigned lower targets than those with smaller ranges.

We first examined the adequacy of existing protected areas for conserving mangroves. We calculated ecosystem services provided by mangroves in protected areas as the sum of the values provided by previous studies[21,29–31]. We then ran three series of prioritisations optimised for the selection of areas that provide substantial benefits in terms of safeguarding biodiversity and associated ecosystem services. These prioritisations were constrained by incremental area budgets (i.e., the maximum area that can be selected) from 1% to 100% of the current mangrove extent globally. This generated a nested set of priority areas for protection, with each planning unit in the analysis assigned a rank depending on when it was selected. Planning units with ranks closer to one represent more important areas to biodiversity and ecosystem services at the global scale. To assess the value of including ecosystem services in the analysis, we then compared the results with a prioritisation optimised for biodiversity protection only. We also ran weighted prioritisations that favoured the selection of areas that provide more ecosystem services over areas that are important for biodiversity but provide fewer services (see "Methods" section). We then analysed the efficiency (i.e., percentage of targets reached and amount of ecosystem services provided per area of mangroves protected) of these prioritisations.

## Results

### Coverage of biodiversity and ecosystem services in current protected areas

The current global system of strictly protected area covers 13.5% of the distribution of mangroves globally (Supplementary Fig. 1). This increases to 43% of the global distribution of mangroves protected when we also include protected areas under IUCN categories V, VI, and

those unclassified (Supplementary Fig. 2). Considering strictly protected areas, the coverage protected varies regionally: greatest protection is in the Americas (24.7%), with less protection in Oceania (11.1%), Asia (10.9%) and Africa (3.5%; Fig. 1a). Currently, 11 countries protect >70% of their mangrove area, including Taiwan (83.0%) and the USA (77.5%), but 67 countries protect <10% of their mangroves, including Vietnam (2.8%) and Thailand (<1%). Considering the top-10 countries with the greatest mangrove area, protection levels were lower than the global mean coverage in Brazil (12.9%), Malaysia (8.8%), India (<1%), Myanmar (<1%), Nigeria (<1%) and Papua New Guinea (<1%); but higher in Mexico (21.2%), Australia (18.0%), Indonesia (16.1%) and Bangladesh (14.5%). Similar results were observed at the continental scale when we considered all protected areas, with greatest protection in the Americas (72.4%) compared to other continents (~30%; Supplementary Fig. 3). However, we observed a large variation at the country scale, with 27 countries that protect >70% of their mangroves and 23 that protect <10%. This is the result of many countries having protected areas that are not strictly protected. For example, there is a large increase in the mangroves considered protected in Brazil, from 12.9% under IUCN categories I-IV to 88.7% under all IUCN protection categories. Further, the mean area of mangrove patches in strictly protected planning units is 36% higher than in unprotected mangrove units. This could be due to the preferential protection of areas with more mangroves, which could potentially miss the protection of small mangrove patches, or to higher deforestation and fragmentation in mangroves outside protected areas.

Mangroves inside existing protected areas provide extensive ecosystem services in different countries: greatest values of property currently safeguarded from flooding are in Taiwan (7.8 billion USD) and Mexico (2.9 billion USD; Fig. 1b); the most people safeguarded from flooding are in Vietnam (0.25 million people) and Taiwan (0.17 million people; Fig. 1c); the greatest carbon is stored in Indonesia (324.4 Tg C) and the USA (82.1 Tg C; Fig. 1d); and the most fishers supported by mangroves are in Indonesia (3.4 million fisher days year$^{-1}$) and Bangladesh (2.2 million fisher days yr$^{-1}$; Fig. 1e).

A variable percentage of each ecosystem service is conserved in the 13.5% of global mangrove distribution that is currently in protected areas (Fig. 1b–e). Coastal property values are well represented in the current system, which protects 24.9% of the total value globally (amounting to 12.0 billion USD), and stored carbon is moderately well represented, protecting 14.5% (980.8 Tg C). However, only 6.8% of the total number of people near mangroves are safeguarded from flooding (0.64 million people), and 6.2% of the total fisheries benefit is protected (11.1 million fisher days yr$^{-1}$). It is clear that the current system of strictly protected areas was not developed to optimise protection across all ecosystem services but avoided areas where large populations live.

## Protection of global mangrove priority areas for biodiversity and ecosystem services

When the current network of protected areas is ignored, global mangrove priority areas identified for protection efficiently maximise the combination of biodiversity and ecosystem services (Fig. 2). These priority areas are broadly distributed (Supplementary Fig. 4a), but most high-priority areas (i.e., top-ranked 10% of mangrove areas) are in Asia (75.4% of the mangrove area; Fig. 2b, e) and Oceania (17.9%; Fig. 2d). This is a consequence of their high mangrove species richness (Supplementary Fig. 5) and high levels of ecosystem services (Supplementary Fig. 6). When we extend the analysis to the mid-priority areas (i.e., top-ranked 30% of mangrove areas), the selection was more-evenly distributed globally, with additional mangroves selected in the Americas (26.4%) and Africa (6.9%). There were also many mid-priority areas in Indonesia, India, Malaysia, Papua New Guinea, Thailand and Australia. This reflects the wealth of ecosystem services provided by mangroves in these

countries and the large areas occupied by mangroves (Supplementary Figs. 4b–e and 6).

Considering priority areas identified for conservation of biodiversity and ecosystem services, only 4.9% of the high priority and 12.1% of the mid priority are within the current protected area system. Thus, the existing protected area system has poor coverage of priority areas (i.e., both <13.5%). More concerning is that the current protection only reaches 6.8% (64 out of 944) of the conservation targets set for individual mangrove species across biophysical typologies and biogeographic provinces (Fig. 3c). None of the species reaches all these individual conservation targets. Therefore, the current system of strictly protected areas neither meets reasonable conservation targets for biodiversity nor efficiently conserves ecosystem services.

There was also extensive variability at the continental and country scale in the efficiency of current protection of priority areas. In the Americas, 30.4% of the high-priority areas are currently strictly protected, in Oceania 8.8%, while in Asia and Africa <2%. A total of 14 countries currently strictly protect >30% of their high-priority areas but 63 countries do not protect any of them. Among the countries that have the largest percentages of high-priority areas, only Taiwan (83.0%) and Australia (21.1%) are strictly protecting >5% of those areas, while India, Indonesia, Malaysia, Myanmar, Papua New Guinea, Thailand and Vietnam strictly protect <5%.

## Expanding protection by adding to the current protected area system

As it is unlikely that current strictly protected areas will be moved, we also analysed their expansion to protect 30% of the global mangrove area[45], as proposed by the "30 by 30" target[4]. We used the same approach as in the initial prioritisation: simultaneously maximising the protection of biodiversity and preservation of ecosystem services (Supplementary Fig. 7). We found that almost all priority areas to expand the system to 30% are in Asia (63.6%) and Oceania (17.0%), highlighting their high conservation value (Fig. 1a). At the country level, the greatest protected area expansion was needed in Indonesia (30.1%), Papua New Guinea (10.5%), India (9.5%), Panama (5.3%) and Thailand (5.0%; Fig. 1a). Similarly, we analysed the expansion of the current system of protected areas (including those that are not strictly protected) to protect 50% of the global mangrove area (Supplementary Fig. 8). Results are similar, with most priority areas for the expansion in Asia (74.5%) and Oceania (19.5%; Supplementary Fig. 3).

Ecosystem services provided by the proposed expansion to 30% of the global mangrove area has the potential to safeguard and sustain an additional: 16.3 billion USD of coastal property value (33.7% of the total amount represented by mangroves globally), 6.1 million people (64.8%), 1173.1 Tg C (17.4%) and 50.7 million fisher days year$^{-1}$ (28.6%; Figs. 1b–e, 3d). Countries that provide most of these additional ecosystem services are India (6.1 billion USD, 2.1 million people, 14.3 million fisher days year$^{-1}$), China (3.7 billion USD), Vietnam (2.9 million people), Indonesia (460.0 Tg C), Papua New Guinea (156.0 Tg C) and Bangladesh (7.0 million fisher days year$^{-1}$; Fig. 1b–e).

## The efficiency of including ecosystem services

The solution of the conservation plan that optimise selection for biodiversity and the one that optimise selection for both biodiversity and ecosystem services were partially overlapping (Cohen's Kappa index of 0.7, Supplementary Fig. 9). Remarkably, the efficiency of solutions meeting conservation targets when optimising for both biodiversity and ecosystem services was very similar to the efficiency of solutions when optimising for biodiversity alone (Fig. 3a, c, e). For example, a similar percentage of targets was reached by the expansion to 30% for the protection of only biodiversity (81.0% of the target) and for biodiversity and ecosystem services (80.0%). Similar results were observed across all area budgets. This seems to be related to higher ecosystem services provisioning in areas of higher biodiversity (here

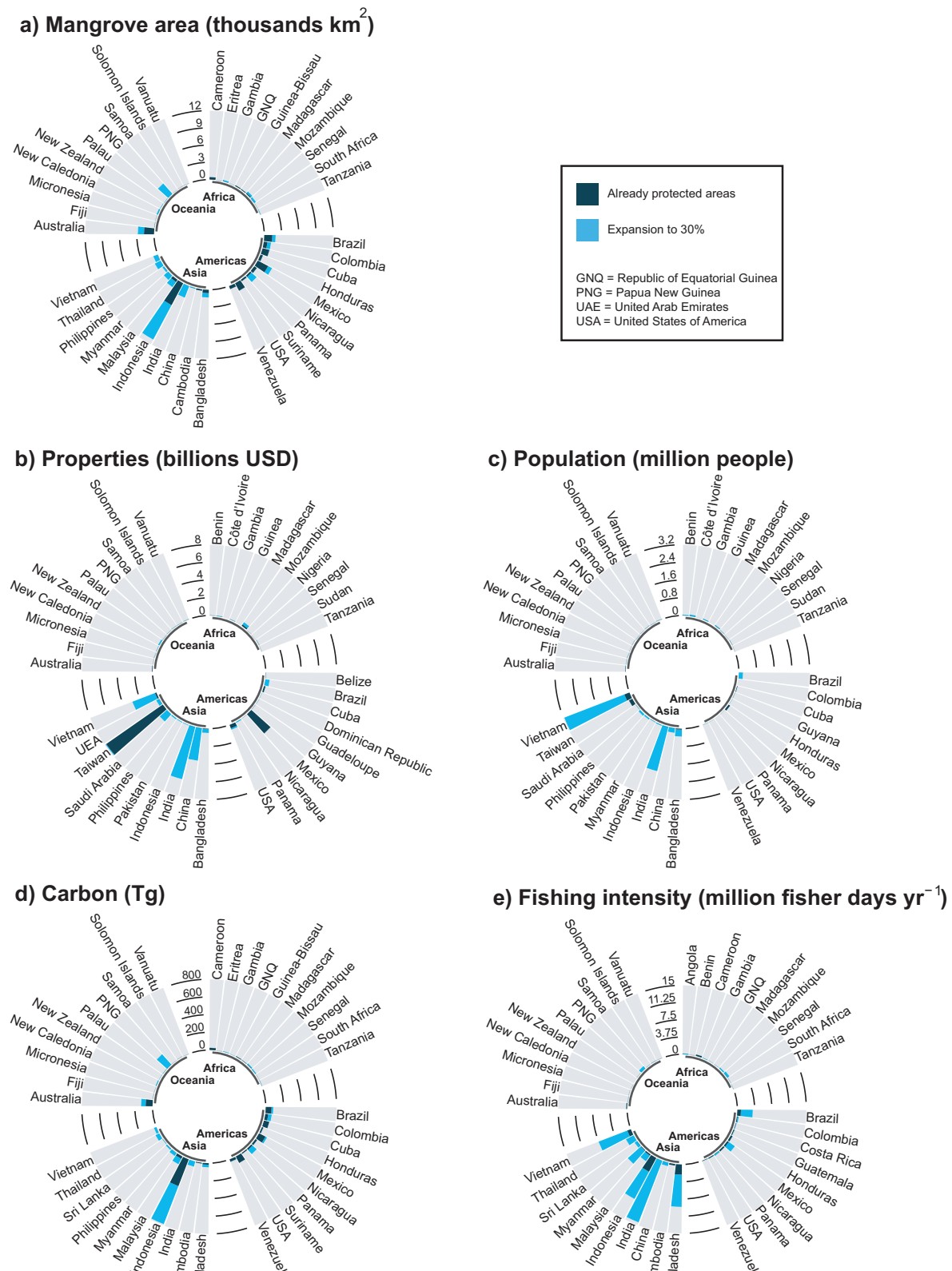

**Fig. 1 | Cumulative mangrove area and cumulative ecosystem services, by continent and country, for current strictly protected areas and for priority areas added to the current system. a** Mangrove area and (**b**–**e**) ecosystem services provided by current strictly protected areas (dark shade) and priority areas to expand the current system to 30% coverage (light shade) resulting from a prioritisation that maximises protection of biodiversity and ecosystem services. Only the top-10 countries per continent are shown: (**a**) by mangrove area and (**b**–**e**) by amount of ecosystem services provided.

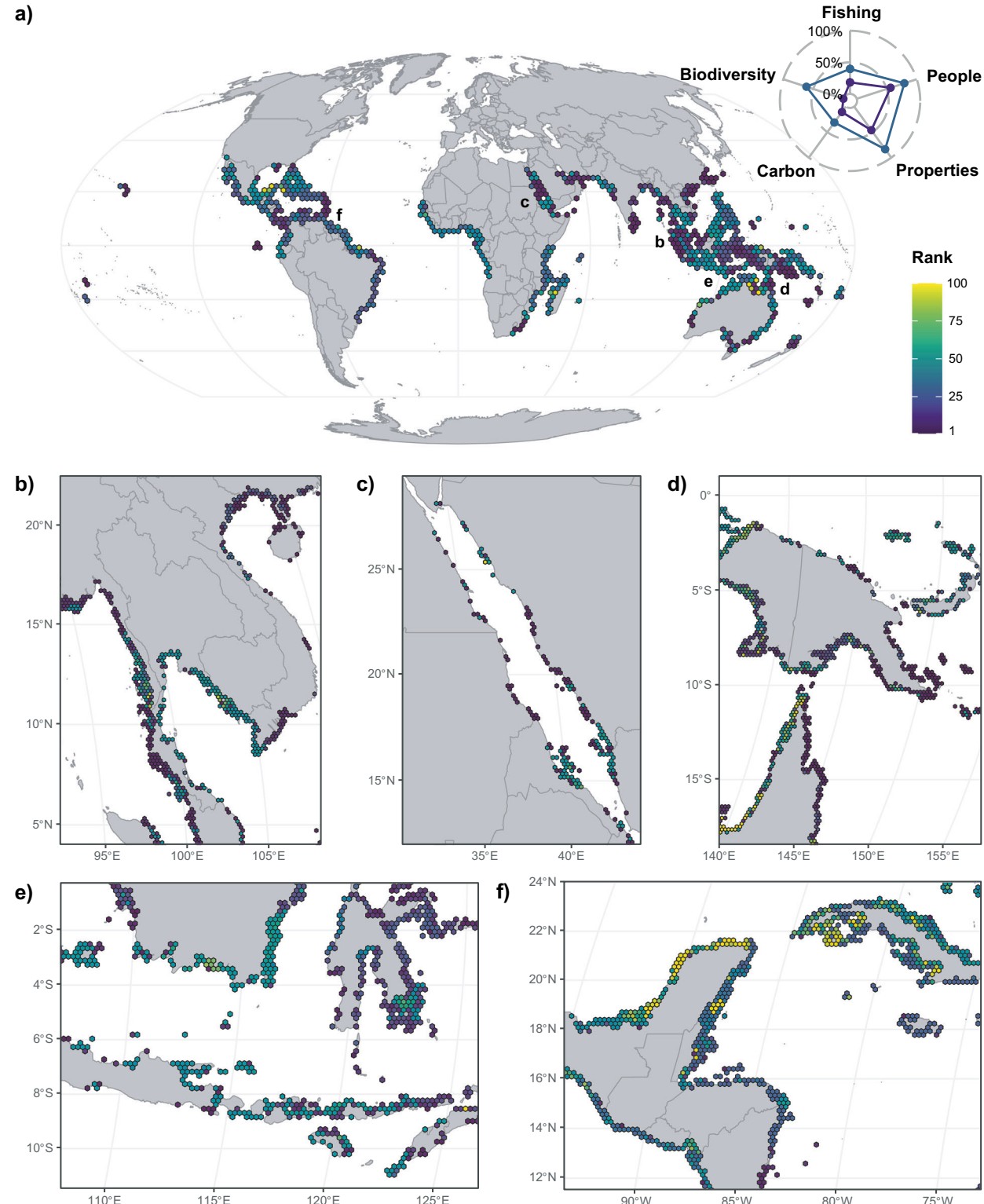

**Fig. 2 | Global mangrove priority areas for protection. a** Priority map, with hexagonal bins aggregated at a resolution of 200-km alongshore for visualisation, shows the median value of the rank of the planning units in the selection (the lower the rank, the more important the planning unit is for protecting biodiversity and delivering ecosystem services). The radar plot shows the percentage of ecosystem services protected and the percentage of conservation targets reached in the top-ranked 10% and 30% priority areas (same colour scale as the map). Inset maps (planning units at the analysis resolution of 20-km alongshore) show results for: (**b**) Southeast Asia; (**c**) The Red Sea; (**d**) West Papua, Papua New Guinea, Northern Australia; (**e**) Indonesia and (**f**) Mesoamerica and Cuba.

estimated by species richness) that drive the selection of mangrove areas that provide more ecosystem services. This could be explained by higher functional diversity in areas of higher species richness, which can guarantee the provisioning of multiple services.

Nonetheless, in the prioritisation that weighed (i.e., increase the probability of selection of a certain feature) the ecosystem services more than biodiversity, we observed an even steeper rise in the services selected at the beginning of the prioritisation (Supplementary

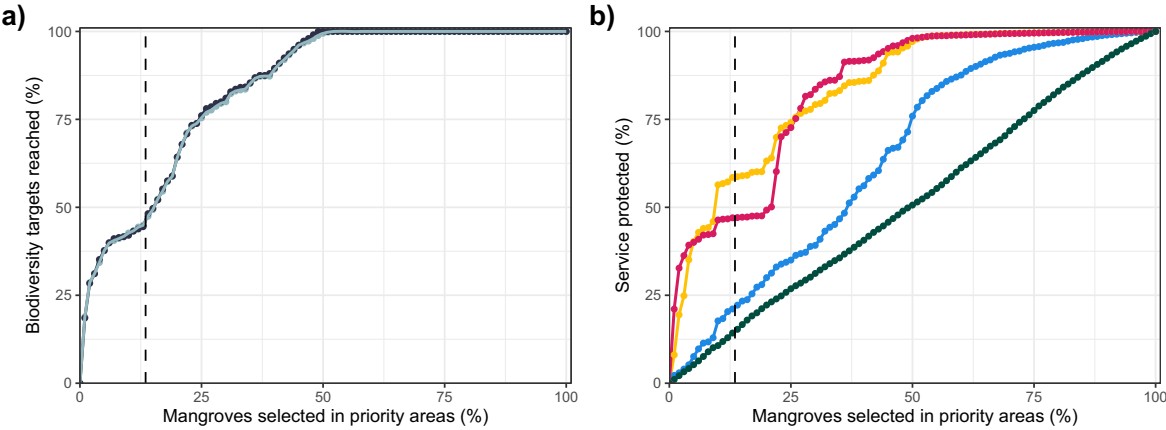

### Selection of priority areas considering all mangrove areas

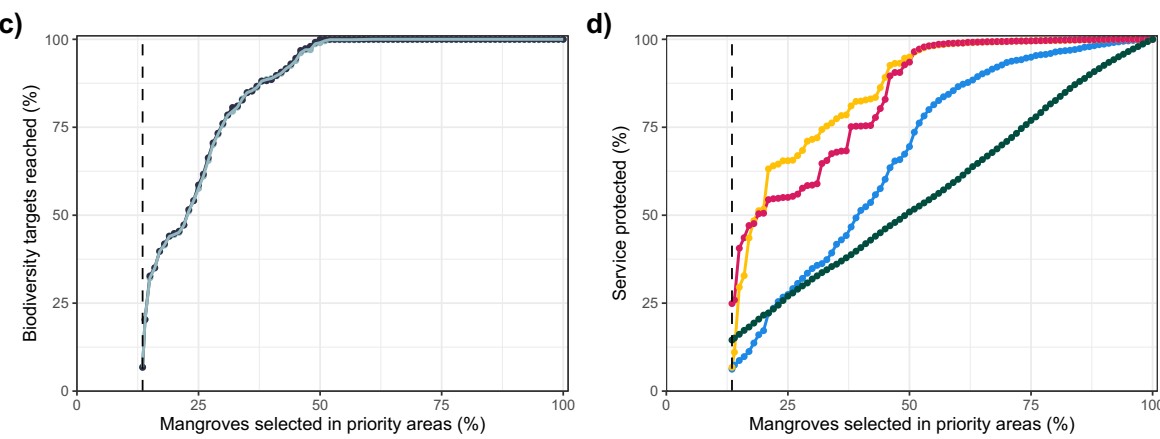

### Selection of priority areas adding to current strictly protected areas

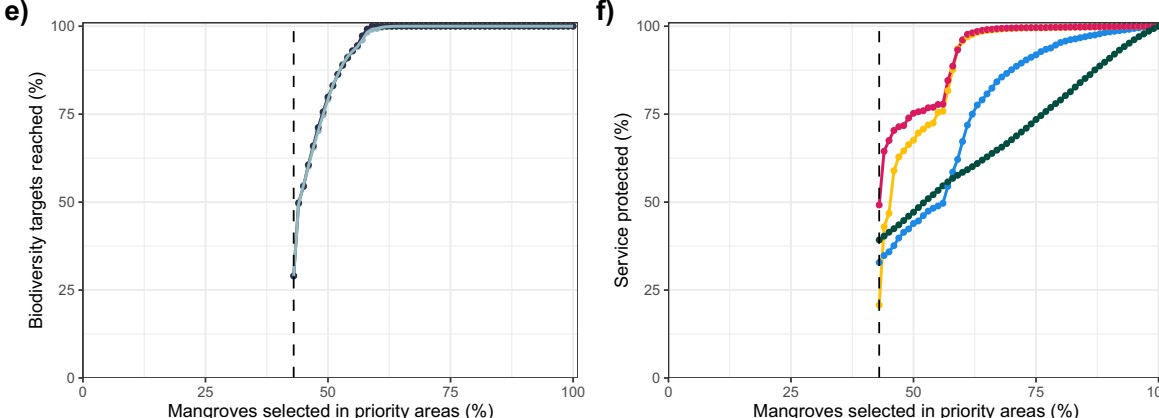

### Selection of priority areas adding to all current protected areas

**Fig. 3 | Conservation targets reached and ecosystem services conserved, in a series of prioritisations based on incremental area budgets.** For incremental area budgets (i.e., increasing the area of mangroves protected): (**a**, **c**) Conservation targets reached (%) in a spatial prioritisation optimised for biodiversity alone or for both biodiversity and ecosystem services; and (**b**, **d**) the percentage of the total of each ecosystem service protected. Prioritisations (**a**, **b**) optimise the selection considering all mangrove areas; (**c**, **d**) add the selection to the current system of strictly protected areas (13.5% of mangrove area, indicated by a dashed vertical line) and (**e**, **f**) add the selection to the current system of all protected areas (43% of mangrove areas, indicated by a dashed vertical line).

Fig. 10). This could also explain why the prioritisation that targets biodiversity only and the one that targets both biodiversity and ecosystem services track each other. For larger weights, the efficiency in the selection of areas of high ecosystem services provisioning increases, but at the expense of the efficiency in protecting mangrove biodiversity (Supplementary Fig. 10). This means that by protecting a similar extent of mangroves, it is possible to reach the same conservation goals, whilst harnessing more ecosystem services. However, we observed variability in the percentage of conservation targets reached when comparing the efficiency of protecting mangrove biodiversity and ecosystem services adding to the current system of strictly protected areas. A much larger percentage of targets (44.9%) are reached when we protect the top-ranked 13% (i.e., equivalent to the current area of the protected area system) of mangrove areas ignoring the current strictly protected areas, compared to the much-smaller percentage of targets reached by the current strictly protected area system (6.8%, Fig. 3a, c). For larger area budgets, the number of targets reached becomes more similar. In fact, when currently strictly protected areas were ignored, 99.5% of the conservation targets (939 out of 944) were reached when 50% of the global mangrove area was selected (Fig. 3a). Similarly, when we started with the current strictly protected areas, 99.0% of the targets were reached when 50% of the mangrove area was selected (Fig. 3c). When protected areas were included in the analysis a larger mangrove area required protection to reach all the conservation targets (Fig. 3e).

Regardless of whether we started with existing strictly protected areas or not, ecosystem services considered were efficiently maximised in the prioritisation (Fig. 3b, d, f). Planning units selected earlier in the prioritisation provided rapid gains in the amount of ecosystem services protected (Fig. 3b, d, f). Diminishing returns were provided by areas selected later, as fewer areas with high values of ecosystem services were available for selection. This is evident for people and properties safeguarded from flooding. The frequency distribution of those services shows that they vary by many orders of magnitude, with

a few areas providing substantial services that could be preferentially chosen early in the prioritisation, but the majority of areas providing low service values (Supplementary Fig. 11a, b). By contrast, the frequency distribution of carbon sequestration shows that it varies by much less than an order of magnitude, with most values in a tight range (Supplementary Fig. 11c), resulting in a near-linear increase in the amount of stored carbon conserved as the protected area system is expanded (Fig. 3b, d, f). Further, after an initial large gain in ecosystem services, some areas are selected that provide few additional ecosystem services because they are important for mangrove species conservation (Fig. 3a, c, e).

We found that mangroves in the current protected areas system provide substantial carbon sequestration and coastal risk reduction for properties, but low amounts of fishing and coastal risk reduction for people (Fig. 4). The mean value of coastal risk reduction for properties, and the mean carbon stored, is similar in planning units currently protected and in those selected by the prioritisation that optimised protection for biodiversity and ecosystem services (Fig. 4a, c). Lower values have been observed when comparing mangroves targeted by less strictly protected areas compared to those selected by the prioritisation. On the other hand, we found much higher mean values of people safeguarded from flooding and fishing intensity in planning units selected by the prioritisation relative to those currently protected (Fig. 4b, d). This is probably because protected areas have been preferentially placed in areas with lower conflicts with fisheries and people, but could also be related to the capability of protected areas to reduce fishing pressure and prevent the presence of people (Fig. 4b, d). The weak correlation between the value of properties and the number of people safeguarded from flooding by mangroves (Fig. 4a, b) is likely to be a consequence of the difference in the value of properties between rich and poor countries. We also found that the prioritisation that optimised protection for both biodiversity and ecosystems yielded higher ecosystem service values than the prioritisation that

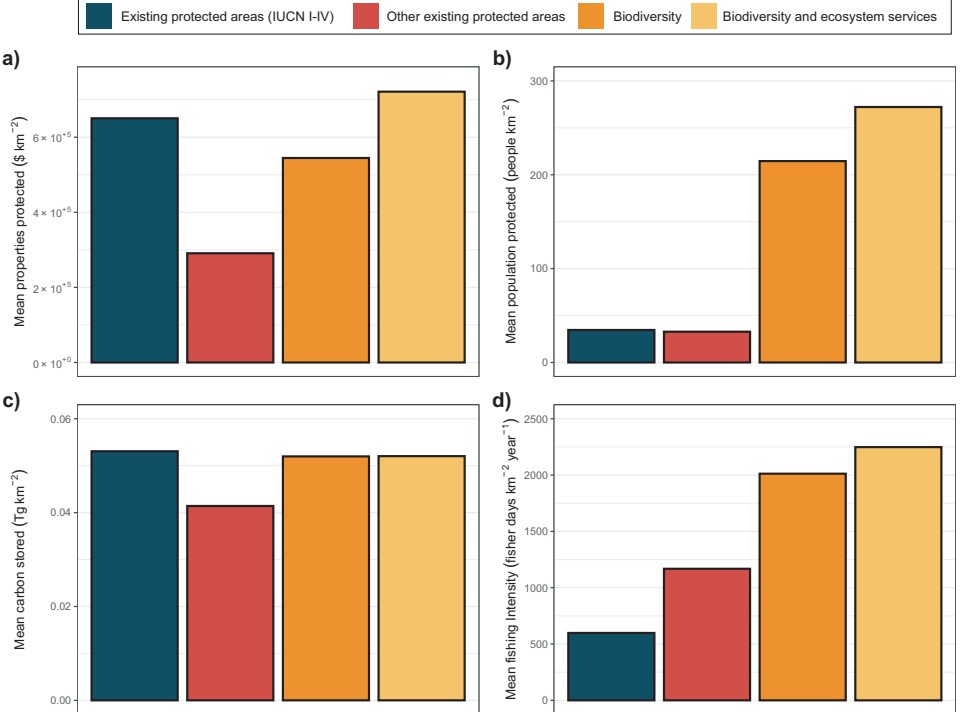

**Fig. 4 | Mean ecosystem services provided by mangroves in different protected areas.** Existing IUCN I-IV protected areas system (blue), other existing protected areas (red), and proposed expansion to 30% for protection for biodiversity alone (orange) and for biodiversity and ecosystem services (yellow). Ecosystem services were: (**a**) properties and (**b**) population safeguarded from flooding, (**c**) carbon stored and (**d**) fishing intensity supported by mangroves.

optimised protection solely for biodiversity, except for carbon storage (Fig. 4). The difference in mean carbon stored between the prioritisations and the currently protected areas is small compared to the large increase in benefits provided by other services because of the near-linear increase in benefits compared with other ecosystem services (Figs. 3b, d, f and 4c and Supplementary Fig. 10).

## Discussion

Given the economic and social costs of conserving biodiversity, areas for protection need to be prioritised. We identified priority areas for protection of mangrove biodiversity and their valuable ecosystem services in all continents, but Asia stands out as the most important. It has large biodiverse mangrove areas that support high levels of fishing and protect many coastal communities[21,31,40]. The current strictly protected area system conserves 13.5% of mangroves globally, but is not representative of all mangrove species, and does relatively poorly in protecting high priority conservation areas in our analysis. For example, the current system protects only 4.5% of the high-priority areas for ecosystem services, and thus performs worse than if areas for protection were selected randomly. This highlights that there is considerable opportunity to optimise future protection. Indeed, most high-priority areas are not currently protected. Some countries, such as Indonesia, India, Vietnam and Papua New Guinea have many high-priority areas, but have low levels of current protection. Considering that most priority areas identified are in developing countries and that some of their ecosystem services provide benefits both locally and globally, we need to support efforts within these countries to implement enhanced management measures to maintain biodiversity and provisioning of ecosystem services in their mangrove ecosystems[46]. High-income countries, whose consumption levels exacerbate biodiversity and carbon storage loss in middle and low-income countries[47], could help finance the global protection of mangroves and alternatives to activities that have a negative impact on biodiversity and their ecosystem services.

In many countries where mangroves thrive, awareness of the intrinsic value of mangroves is increasing and with it, their protection (e.g., increasing mangrove conservation in Brazil[48]) and restoration (e.g., the Indonesian government promising to restore 600,000 ha of mangroves by 2024[49]). Considering that the restoration of mangroves is expensive and not always effective[50,51], the protection of natural mangroves is a more effective conservation solution in areas important for biodiversity and ecosystem services. Further, restored mangroves provide lower ecosystem services than natural and naturally regenerated[50,51]. Multiple countries and regions are currently engaged in coastal conservation planning to expand their protected areas system, and some are trying to include ecosystem services (e.g., the Caribbean Challenge Initiative[52], The Seychelles Marine Spatial Plan Initiative[53] and Belize Nationally Determined Contribution[19]). Although there are multiple examples in the literature of conservation planning that include ecosystem services at a regional scale[15-17], almost all initiatives calculate the values of services as co-benefits and do not include them in the process for the selection of protected areas. Our results show that including ecosystem services in the selection of areas for protection provides benefits and can be considered when implementing new protected areas systems. The patterns we found are similar to those reported by Arkema et al.[19]. Increases in both fisheries benefits and coastal risk reduction become small for high conservation targets, while carbon sequestration benefits continue to grow linearly. This seems to be related to the non-linear relationship between fisheries benefits and coastal risk reduction with mangrove area[19]. Countries in the analysis where large areas were selected as high priorities for conservation are obvious places for expansion, building on the encouraging results that have been observed in multiple countries where protection is increasing[54]. The identification and protection of the areas providing greater ecosystem services can also be used to

incentivise the enforcement of protected areas, highlighting the benefits to people. For example, our results could inform an expansion of protected areas in high-priority areas in Asian countries such as Myanmar, Malaysia and Vietnam, where we found few protected areas and where mangroves are threatened by deforestation for aquaculture and agriculture[55].

We also found that many priority areas are close to urban centres where there are higher ecosystem services provided by mangroves. In these areas, mangroves are most at risk from clearing, fragmentation, coastal squeeze and changes in water quality and sedimentation[56]. Although the need for conservation is evident, implementation is not always straightforward. Future research could also include spatial patterns of loss to highlight areas at higher risk and therefore require swift protection. Enforcing protected areas can cause conflicts with local communities and industries[57]. To minimise these conflicts, conservation planning generally avoids areas of intense human activity, minimising the opportunity cost[5]. Thus, protected areas are often established in places that avoid high human populations, valuable industries, rich agricultural land and productive fishing grounds[8,9]. In selecting areas for protection by minimising the opportunity cost, these planning processes can miss the most valuable biodiversity hotspots and are blind to the ecosystem services provided by these areas.

By contrast, we identified areas for the implementation of conservation measures that maximise the protection of ecosystem services provided by mangroves and thus have large human benefits. We identified areas that provide extensive ecosystem services but also meet conservation targets for the protection of species and provide direct services to local people. This approach complements those that focus on the economic benefits of biodiversity protection rather than producing a spatial plan (e.g., Costanza et al.[6]) and those that identify ecosystem services co-benefits secured by focusing on biodiversity protection (e.g., Sievers et al.[13]). Although our approach will help ensure the sustainable delivery of those benefits into the future, it might pose challenges because some of the proposed protection, close to urban areas, could be difficult to manage due to human pressures[58,59]. Currently, >50% of protected mangrove ecosystems suffer high human pressure and are threatened by conversion, fragmentation and degradation[59]. Further, anthropogenic activities are the main cause of ineffective protected area management[60]. However, this should not eliminate the need to protect these areas, which could be habitats for important species[58] and provide greater ecosystem services exactly because of the anthropogenic activities they support or protect.

A shift might be needed in how many spatial prioritisation problems are approached. Including ecosystem services in spatial prioritisation could be considered for systems where the link between services and biodiversity is clear. For example, there is strong evidence that the protection of biodiversity enhances ecosystem services such as coastal protection, carbon sequestration and enrichment of fisheries in many coastal ecosystems such as coral reefs, salt marshes and seagrasses[23,61-63]. In these cases, the enhanced management of these systems (including through implementation of protected areas) can be mitigated by the benefits provided by the safeguarded ecosystem services. In other systems, including many offshore marine habitats, the link between biodiversity and ecosystem services might be less direct, and minimising conflict with industry such as fishing might be appropriate.

Our global analysis has several caveats that should be considered. First, we focused on benefits provided by ecosystem services, but we did not quantify the cost of protecting selected mangrove areas. Future analyses could measure the trade-off between costs and benefits of protection. Second, although data used on ecosystem services and species distributions are the best available, they are relatively coarse. Considering that global prioritisation can hardly adequately

represent fine-scale variation in the data or local actions[64], the development of systematic conservation plans at local scales are necessary for the effective implementation and management of protected areas[5]. Third, as boundaries of protected areas do not always line up with the boundaries of our planning units, we assigned a threshold of 50% of the area of mangroves in a planning unit to consider it protected, as has been adopted elsewhere (e.g., Runge et al.[65]). However, for studies focused on management, it could be defined based on the goal of the prioritisation. Therefore, our results should be interpreted as highlighting areas that could be considered for future analysis. Fourth, in our study, including the coastal risk reduction and fisheries benefits provided by mangroves has favoured the selection of mangroves close to urban areas. However, larger benefits for biodiversity could be provided by protected areas distant from urban areas, considering the multiple pressures for biodiversity related to human activities and infrastructure[66,67]. We used data from the best available dataset on fishing intensity in mangroves[31], but we recognise that the results could differ using catch or another metric that combines both supply and demand. In future studies, other metrics of fisheries productivity could be included to evaluate if the fishing intensity and catch overlap. Fifth, all ecosystem services datasets (i.e., Sanderman et al.[29]; Simard et al.[30]; Menéndez et al.[21]; Zu Ermgassen et al.[31]) do not consider that different mangrove species and assemblages provide different benefits[68] and that cryptic ecological degradation could result in the loss of ecosystem services provisioning[68,69]. Sixth, data from Zu Ermgassen et al.[31] did not consider protection status when estimating fishing intensity in mangrove areas. This could have potentially overestimated the benefits for fishing in areas that are no-take marine protected areas. Last, we only considered protected areas, but alternative approaches such as other effective area-based conservation measures, indigenous protected and conserved areas, communitybased conservation and payment for ecosystem services could also ensure positive biodiversity outcomes that are equally valid and help meet societal goals[70-73].

Expanding the current protected area system for mangroves by optimising protection for both biodiversity and ecosystem services could have large benefits, particularly in terms of coastal protection for infrastructure and people, and supporting fisheries. We found that biodiversity objectives could be met whilst simultaneously maximising ecosystem benefits, with little or no increase in protected area required. A protected area system predicated on the idea of maximising benefits rather than minimising opportunity costs to people would not only be valuable for livelihoods of coastal people, but also help address global threats such as climate change and biodiversity loss. We hope that this work can inform policymakers and other stakeholders working in biodiversity conservation about the importance of explicitly considering ecosystem services in conservation decision making.

## Methods
### Study area
Our study area encompassed the global distribution of mangroves (137,600 km² in size). This spatial extent was identified using the 2016 version of the Global Mangrove Watch (GMW) dataset[45] (https://www.globalmangrovewatch.org; Supplementary Fig. 12). We subdivided the study area into 9111 hexagonal planning units with a spatial resolution of 20 km alongshore. Datasets included in this analysis have spatial resolution ranging from 25 m (i.e., global mangrove distribution from Bunting et al.[40]) to 20 km (i.e., coastal risk reduction values from Menéndez et al.[21]). Hence, we standardised all datasets to a (20 km along the coast × 20 km inland to include mangroves that extend inland ~ 400 km²; Menéndez et al.[21]). Higher resolution data on mangrove species and ecosystem services are not available. For example, Chaplin-Kramer et al.[74] report data on the number of people safeguarded from flooding by mangroves (and other habitats) at a much

coarser 1° resolution (~12,300 km² at the equator). The use of relatively coarse 20-km alongshore planning units in the analysis also assists in the protection of mangrove patches that are large enough to ensure the provision of services[75]. Planning units were created under the Mollweide equal-area coordinate reference system (ESRI:54009), and all spatial datasets described hereafter were reprojected to this coordinate system prior to analysis. Global coastline data was retrieved using the rnaturalearth package[76]. This data was also used, combined with Exclusive Economic Zones boundaries[77], to aggregate the results by country and continent. All analyses were performed using the R statistical computing environment (version 4.2.1)[78].

### Mangrove biodiversity
We used a variety of data to describe the spatial distribution of mangrove biodiversity. First, we obtained geographic range data of the 65 most-common mangrove species from the IUCN Red List of Threatened Species[79]. These scale-independent vector data report only the broad range of distribution of the species. Similar to Brooks et al.[80], we intersected these ranges with GMW mangrove distribution data to produce high-resolution species distribution data (25-m resolution). Second, we obtained mangrove biophysical typology data from Worthington et al.[42]. These data classify mangroves according to geomorphic features (i.e., deltaic, estuarine, lagoonal and open-coast mangroves) and sedimentary settings (i.e., carbonate and terrigenous mangroves). Each mangrove species distribution was divided into separate features by intersecting it with these typologies. Third, we obtained marine province data from Spalding et al.[43]. These data classify the world's coasts and shelves into 12 realms, 62 provinces, and 232 ecoregions based on the analysis of reviews of publications and expert advice. Each planning unit was assigned a province that predominantly spans the area.

### Conservation features
Three types of data were included in the analysis to ensures the greatest benefits for biodiversity representation, namely: (1) mangrove species distribution; (2) mangrove biophysical typology and (3) and marine provinces of the world. Data on the distribution range of mangrove species were included to ensure a minimum area requirement for each species in each mangrove biophysical typology and marine province, for the long-term protection of the species. Following Rodrigues et al.[44], the conservation target for each species was calculated using a $\log_{10}$-interpolation between a maximum target (100% of the distribution for species with a range <10,000 km²) and a minimum target (10% of the distribution for species with a range >250,000 km²). This choice tries to counterbalance the bias towards the larger representation of more widespread species when using a constant representation target. In fact, species with a smaller distribution range require more-demanding conservation targets, considering they tend to be less abundant and thus more vulnerable to natural events and anthropogenic activities[44]. Although this approach of setting targets is somewhat arbitrary, it is common in multiple studies (e.g., Runge et al.[65], Hanson et al.[81]) and is used by international organisations (e.g., Claes et al.[82]). In local studies, other methods to set the targets, such as the use of fixed area-based policy targets[83], minimum viable population analysis[84] and expert-based thresholds[10], could be used. Including these features ensures the protection of the heterogeneity of mangrove communities across biogeographic habitats[85]. Each data layer was intersected with the mangrove distribution, resulting in a biodiversity value for each planning unit that represents the total area of mangroves that intersect that feature in the planning unit. A full list of all the biodiversity data is in Supplementary Table 1.

### Ecosystem services
We included ecosystem services (Supplementary Fig. 6) in the analysis because it allows the selection of protected areas not only based upon

meeting biodiversity objectives but also ensuring social and economic benefits. Mangrove ecosystem services considered were coastal protection of both people and properties, carbon sequestration, and benefits for fisheries. Coastal protection and benefits for fisheries were included in the analysis because they are important in supporting livelihoods of coastal people[20,31]. In fact, mangroves coastal risk reduction has a direct impact on the lives of coastal communities[20], whilst protecting areas to increase benefits for fisheries can ensure food security for coastal populations[86]. Carbon sequestration was included because it considers the benefits provided by helping to stabilise greenhouse gas levels and limiting effects of climate change[87]. Since some of these datasets were based on different underlying mangrove datasets, we interpolated values from each of the datasets to our planning units using the nearest neighbourhood method. Specifically, after intersecting the datasets with the planning units, those with missing values were assigned the value of the nearest planning unit with data. Other methods are available (e.g., bilinear interpolation used in Sala et al.[12]), although they all have their own assumptions. When higher-resolution spatial data on biodiversity distribution and ecosystem services are available, our analyses could be improved. We only included three of the multiple ecosystem services provided by mangroves. Future research, that uses the same method we applied here, could focus on evaluating and including other services, such as tourism, when data become available. For all ecosystem services included in the analysis, spatial interconnection between production and delivery of the services was not considered given the local benefits that all these services ensure (i.e., local coastal risk reduction, profits from the sale of seafood and carbon credits).

**Coastal protection.** The number of people and the value of the properties (USD) safeguarded from flooding by mangroves per 20 km of coastline were obtained from Menéndez et al.[21]. These data were calculated by comparing flood impact scenarios with and without mangroves, combined with the global distribution of people and properties and a corresponding damage function, using the Philippines as a pilot case. Offshore and nearshore dynamics generated by tropical cyclones were simulated in the Philippines and then validated using real data. The results were then used to create a predictive model that used oceanographic variables to predict cyclone parameters. Another model was produced to analyse the role of mangroves in nearshore dynamics. First, offshore hydrodynamic conditions were resolved under both regular climate and tropical cyclones conditions. Then, nearshore dynamics were obtained by propagating regular climate using hybrid downscaling. For tropical cyclones, nearshore dynamics were produced through the regression model derived from the Philippines results. The role of mangroves in attenuating floods was included to obtain the results for scenarios with and without mangroves. Finally, flood maps and an assessment of the consequence of the flooding on people and industrial and residential stock were produced. The caveats listed in Menéndez et al.[21] are related to the limitations of bathymetry, topography and mangroves data, and of the models of the two-dimensional propagations of nearshore waves and storm surges. Further, some countries with <100 ha of mangroves or where benefits were too high (>USD 50,000) were removed from the analysis. In our study, we ran the analysis for mangroves in those countries, but the quantitative results on the amount of coastal protection provided should not be considered. The value for each planning unit was assigned using the nearest neighbour method. Values were then transformed to risk reduction per km$^2$ of mangroves by dividing the number of people and the values of property safeguarded from flooding by mangroves, as reported by Menéndez et al.[21], by the total mangrove area. This area was defined as the total area of mangroves in all planning units that shared the same value of coastal risk reduction after applying the nearest neighbour method (i.e., one planning unit that intersected the data point from Menéndez et al.[21]

plus the closer planning units that were missing a value before applying the nearest neighbour method). For each planning unit, the total number of people and properties safeguarded from flooding was then calculated by multiplying the value per km$^2$ of mangroves by the area of mangroves in that planning unit.

**Carbon storage.** For each planning unit, the total carbon stock was calculated as the sum of mean aboveground carbon and soil carbon in Tg km$^{-2}$ (millions of tonnes of C). We used the estimated aboveground biomass of mangroves from Simard et al.[30]. They calculated the global distribution of aboveground mangrove biomass, linking Light Detection and Ranging (Lidar) heights from the ICESat/GLAS Spaceborne Lidar mission with the Shuttle Radar Topography Mission (SRTM) 30-m resolution global Digital Elevation Model (DEM) of mangrove basal area weighted height. Field data of basal area weighted height were interpolated with the height data provided by SRTM DEM. The results were then used to produce a global basal area weighted canopy height dataset. These data were finally transformed into aboveground biomass measurements using an allometric equation derived from the field data. Soil carbon data were obtained from a machine learning-based model of organic carbon density by Sanderman et al.[29]. They first compiled a mangrove soil carbon database from peer-reviewed literature, grey literature and unpublished data. Then, soil carbon was estimated as a function of depth, an initial estimate of the 0–200 cm organic carbon stock from the global SoilGrids 250-m model downsampled to 30 m resolution[88], and a suite of spatially-explicit covariate layers (i.e., vegetation characteristics, digital elevation data, long-term mean monthly sea surface temperature, tidal elevation amplitude, mean monthly total suspended matter and mangrove typology). The organic carbon density at different depths was then cumulatively summed to calculate organic carbon stock in each pixel where mangroves were identified. Both studies outlined their limitations. Simard et al.[30] indicated the possible effect of using different mangrove extent maps on their findings, while Sanderman et al.[29] indicated the possibility of errors and inaccuracy in the database used and the limited information reported about plot locations. The mean value of aboveground biomass and soil carbon for each planning unit was calculated as the mean of all the values of aboveground biomass and soil carbon overlapped by the planning unit.

**Fisheries benefits.** Mean fishing intensity was used as a proxy for the benefits of mangroves to fisheries. Fishing intensity from zu Ermgassen et al.[31] was calculated as the mean number of fisher days km$^{-2}$ year$^{-1}$ for each planning unit. zu Ermgassen et al.[31] developed a conceptual model that calculated mangrove fishing intensity using data on the number of mangrove fishers (commercial and non-commercial small-scale inshore fisheries), local non-urban population, accessibility to markets, and catchability as a result of the weather. These variables were identified following an expert-based, anonymous, three-round Delphi method. This method was used to identify the variables gathering consensus across multiple experts that were required to score the importance of each factor in determining mangrove fishing intensity and the certainty in their assessment. The first round was based on an open-ended survey where experts were asked to rank the twenty most important factors determining the volume of fish and invertebrate catch. In the second and third rounds, the same experts were asked to score the importance of the factors (1 = not important to 5 = extremely important) about the fishery type they were most familiar with. Factors that had scores of 1 or 2 by >70% of the experts were removed. A total of 16 factors were selected, but only four had suitable data for the development of the map: number of mangrove fishers, local non-urban population, accessibility to markets and catchability as a result of the weather. The number of small-scale fishers, assumed to be proportional to the local non-urban population, was obtained, on a country level, from a literature review. A "market

effect" was included by doubling the number of fishermen per capita for all the areas within 3 h from cities with >50,000 people. Then, the resulting fishermen were assigned to areas of shallow shelf, coral reefs and mangroves based on distance and the ratio 10:10:1. To calculate the number of fish days, zu Ermgassen et al.[31] assumed that fishermen are out at sea 60% of the days. A storm index was calculated to correct the results, considering a 0.75 loss per storm day. This study reported some caveats related to the limited subset of the identified factors that were mappable, the lack of experts with experience in West or Central Africa, the inability to include other structured habitats other than the shallow shelf and coral reefs habitats, the application of global datasets for accounting the impact of storms, and the exclusive focus on near-shore fisheries. Further, using fishing intensity could be misleading because it does not consider the supply of the service considering that the amount of catch could differ among different areas for the same number of days the fishermen are at sea. However, we decided to use these data because, to our knowledge, these are the best available quantitative data on fisheries in mangroves. The study could be improved with finer-scale data when they are available.

### Existing protected areas

Protected area data were obtained from the World Database on protected areas (WDPA)[89]. Data were then cleaned and prepared for analysis following best practices[90,91]. Planning units with at ≥50% coverage were subsequently treated as protected. In most analyses, we concentrated on categories I-IV because there is uncertainty in the effectiveness of protected areas having lower IUCN categories because they might be "paper parks" with no effective governance[92]. This is unrelated to the involvement of indigenous people in conservation that has been proven to be valuable[70–72]. This is also proven by the current conversation in the development of new categories to define different types of protected areas[93].

### Spatial prioritisation

Priority areas for mangrove protection were identified using spatial prioritisation[5]. To avoid numerical issues, all ecosystem services values were rescaled to 0–1000 before prioritisation. A first prioritisation based on currently protected areas was generated without locking in planning units where mangroves were protected. In a second prioritisation that accounted for existing protected areas, planning units that contained mangroves predominantly covered by existing protected areas (≥50%) were locked in. The cost of including a planning unit in the resulting system was defined as the mangrove extent in the planning unit. Following Jung et al.[14], prioritisations were generated using a series of incremental budgets from 1% to 100% (or 14% to 100% when expanding the current system of strictly protected areas and 44% to 100% when expanding considering all protected areas) of the current total extent of mangroves to identify a nested set of priority areas. For a given area budget, the prioritisation process selected areas that minimise the overall shortfall for the conservation targets described earlier whilst minimising the cost (i.e., area). To maximise ecosystem services provided, we also included each as a feature with a conservation target of 100%[10]. To compare how much more expensive in terms of extra area needed our prioritisation based on protecting biodiversity and ecosystem services would be than a more standard prioritisation approach of conserving biodiversity alone, a prioritisation was also generated without ecosystem services. The degree of agreement between the results was calculated using the Cohen's Kappa Coefficient. To explore uncertainty in our analyses, prioritisations were also run for incremental ecosystem services weights (i.e., 10, 100, 1000) to favour the selection of planning units that provide benefits for the protection of ecosystem services over planning units that provide benefits for biodiversity representation.

Higher weights favour the representation of areas that provide more ecosystem services over areas that are important for biodiversity. Weights used here are indicative and arbitrary, and further research in the effects of weighting ecosystem services could be carried out for analysis at the management level to make the best conservation decisions. All prioritisations were completed using the *prioritizr* R package[38] and Gurobi[94], using an optimality gap (the acceptable deviance from the optimal objective) of 0.01%.

### Reporting summary

Further information on research design is available in the Nature Portfolio Reporting Summary linked to this article.

## Data availability

Datasets generated analysed during the current study are available online. The global map of mangroves is available for download from the Global Mangrove Watch[40] website (https://www.globalmangrovewatch.org/). IUCN distribution of mangrove species[41] is available at: https://www.iucnredlist.org/resources/spatial-data-download. The global biophysical mangrove typology[42] and the marine provinces of the world[43] are available for download from the Ocean Data Viewer (https://data.unep-wcmc.org/). The ecosystem services layers can be downloaded for aboveground carbon[30] (https://daac.ornl.gov/cgi-bin/dsviewer.pl?ds_id=1665), soil carbon[29,95] (https://zenodo.org/record/1469348#.Yv9HDcjBy3A), and coastal protection[21] (https://osf.io/ecs4p/). For fishing intensity data[31], please contact the authors of the original article (https://doi.org/10.1016/j.ecss.2020.106975). Data on protected areas can be downloaded from the WDPA database[39] (https://www.protectedplanet.net/). Global coastline data can be downloaded using the rnaturalearth package[76]. Exclusive Economic Zones boundaries[77] are available at: https://www.marineregions.org/sources.php#marbound. Source data are provided with this paper. Datasets generated during the current study are currently available at[96]: https://doi.org/10.5281/ZENODO.8272951. Source data are provided with this paper.

## Code availability

Code to run the analysis in this study is currently available at: https://github.com/AlviDab/SpatialPrioritisation_MangrovesEcosystemBenefits. The code is archived in a Zenodo digital repository[96]: https://doi.org/10.5281/ZENODO.8272951.

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

## Acknowledgements

A.D., K.C.V.B. and F.D.G. were supported by the Erasmus Mundus Joint Master Degree in Tropical Biodiversity and Ecosystems – TROPIMUNDO, which is funded by the European Commission. J.O.H. was supported by Environment and Climate Change Canada (ECCC) and Nature Conservancy of Canada (NCC).

## Author contributions

Conceptualisation by A.D., F.D.G., D.C.D., J.D.E., C.E.L., J.O.H., K.C.B., S.N. and A.J.R.; Methodology by A.D. and A.J.R.; Formal analysis by A.D. and J.D.E.; Writing – original draft by A.D.; Writing – review & editing by A.D., F.D.G., D.C.D., J.D.E., C.E.L., J.O.H., K.C.B., S.N. and A.J.R.

## Competing interests

The authors declare no competing interests.
