## [Peer Review File · Nature Communications]

Reviewers' comments:

Reviewer #1 (Remarks to the Author):

This work spatially identifies areas of significant ecosystem services provided by mangroves. Similar work has been done identifying the value of coastal wetlands for storm protection.

Robert Costanza, Sharolyn J. Anderson, Paul Sutton, Kenneth Mulder, Obadiah Mulder, Ida Kubiszewski, Xuantong Wang, Xin Liu, Octavio Pérez-Maqueo, M. Luisa Martinez, Diane Jarvis, Greg Dee (2021) The global value of coastal wetlands for storm protection, *Global Environmental Change*, Volume 70, ISSN 0959-3780, <https://doi.org/10.1016/j.gloenvcha.2021.102328>.

(<https://www.sciencedirect.com/science/article/pii/S0959378021001072>)

This paper is solid. I don't have much to critique. My only suggestion is perhaps a brief 2-5 sentence exploration of the spatial issue of ecosystem service value relative to the proximity of services to built up areas. Some values (e.g. biodiversity) might be more important further from built up areas whereas other values manifest because of their proximity to built up areas (e.g. storm protection services).

This paper explores the 'triage' ideas of prioritization of biodiversity and ecosystem service benefits. I am very concerned about sins of omission when conducting this sort of analysis. My take on the Ecosystem Services Valuation approach is to help civilization recognize that nature's contributions to people are so large that we must make much more significant efforts to protect nature. This paper suggests that current prioritization strategies focus on mangroves in asia and this analysis will broaden our ideas as to what needs to be preserved. I applaud this conclusion; however, the adoption of cost-benefit analyses that mimic the methods of economists are prone to the same potential errors of failing to account for all values. Nonetheless, existing literature on the value of ecosystems services emanating from mangroves already suggests that mangrove ecosystems should have high priority for protection (Costanza et al., 2014), preservation, and stewardship. Arguably, this prioritization should not be based solely on economic arguments but economic arguments do support the conclusion.

Robert Costanza, Rudolf de Groot, Paul Sutton, Sander van der Ploeg, Sharolyn J. Anderson, Ida Kubiszewski, Stephen Farber, R. Kerry Turner (2014) Changes in the global value of ecosystem services, *Global Environmental Change*, Volume 26, Pages 152-158, ISSN 0959-3780 <https://doi.org/10.1016/j.gloenvcha.2014.04.002>.

Reviewer #2 (Remarks to the Author):

In their manuscript entitled “Priority areas to protect mangroves and maximise ecosystem services”, Dabala and colleagues conduct an optimization analysis of biodiversity and ecosystem service benefits of mangroves. They use the analysis to explore the ecosystem service benefits provided in protected areas currently and to identify priority countries and regions in which to target mangrove protection to achieve both biodiversity and ecosystem service benefits in the future. The paper addresses an important topic that deserves more attention – how to target investments in coastal habitat conservation to achieve ecological and societal benefits. However, I find three major issues with this paper that I think preclude its publication in its current form in Nature Communications. The first two can be addressed with major revisions, but the third is more fundamental to the analysis and without more details on the methods, it’s unclear to me yet whether this can be resolved.

Major comments:

1) Further address the value of global prioritization analyses for conservation

a. I commend the authors for acknowledging the recent exchange in the literature between Chaplin-Kramer et al and Wyborn et al on the utility / limitations of global prioritization analyses. However, given this discourse the authors should do more to connect the results of their paper to national/regional actions that could be taken.

b. For example, the Discussion lacks any specific examples and fails to contribute new insights to the literature. Many of the topics are generally addressed in Sala et al. 2021. The discussion of Asia needing to conserve mangroves is less helpful because actions are rarely taken at a continental level. Can you provide examples of spatial planning in other regions moving in this direction? Or discuss seeds of progress in countries in Asia that could be built upon with this information?

c. The authors perpetuate the lack of attention to local or regional examples by only citing global papers in lines 50-60. Missed opportunity to cite examples of spatial planning and ecosystem services informing decisions or being applied at regional scales relevant to management:

- White et al. 2012 Ecosystem service tradeoff analysis reveals the value of marine spatial planning for multiple ocean uses. PNAS

- Lester et al. 2013 Evaluating trade-offs among ecosystem services to inform marine spatial planning. Marine Policy

- Arkema et al. 2015 Embedding ecosystem services into coastal planning leads to better outcomes for nature and people. PNAS

- Many countries and regions around the world are engaged in coastal spatial planning or aiming for conservation targets that consider ecosystem services (e.g., the Caribbean Challenge Initiative; ocean and marine spatial planning in the Seychelles etc.) what about discussing some of these?

2) Need more information about the methods for calculating each ecosystem service, the optimization analysis, and the targets

a. Targets

- Lines 103-104 – provide more explanation/justification for your approach to developing targets in the intro and methods.

- Lines 182-183 – your description of the radar plot seems to use conservation targets to include biodiversity and ecosystem services. Is that correct? If so, ms needs a clearer explanation of what is meant by targets in the paper (from the intro I thought it was just related to biodiversity) and consistent use of the terminology throughout. Similarly, define what is meant by efficiency in this paper. If you are going to use 944 as a reference point for maximum targets met you need to provide a clear explanation of what that 944 means.

b. Ecosystem services

- There is nothing in main ms about your approach to quantifying ecosystem services. I was expecting a quantitative modeling approach. In the introduction, include in the overall approach that you are using existing published data layers for the current magnitude of ecosystem services so the reader isn't expecting new modeling/assessment.

- Need to give substantially more information about how ecosystem services were estimated for each input dataset into the analysis. The reader shouldn't be expected to read several other papers to understand the information that went into the analyses in this paper and the assumptions for each service.

- For example, did the Mendez paper account for differences among mangrove taxa or other habitat characteristics in quantifying the coastal protection service? Need more information about nearest neighbor method. Does your analysis take into account the spatial interconnections between service production and delivery? If so, how?

- Similarly, there are major limitations in using fishing effort data. What fisheries are included/excluded from this data layer? How will fishing effort be redistributed with MPAs? How do different taxa, morphological characteristics taken into account in the biodiversity part of the analysis relate to fisheries benefits in the ecosystem service assessment?

- How does management in the existing MPAs and conservation interventions relate/influence ecosystem services? Do the existing protected areas prohibit impacts to mangroves (e.g., harvesting, development)? Do they prohibit fishing? If the MPAs prohibit fishing, how is this handled in the analysis?

c. Optimization

- Need more explanation of lines 241-245. This is the first time that the ms mentions “weighting.” More explanation may be in the methods, but if you are going to mention weighting in the results/discussion then you have to introduce it earlier in the ms and give the reader some idea of how weighting fits in. What is meant by “In analyses”? Which analyses?

- Did you use the weighting to explore different societal preferences for combinations of services and biodiversity along an efficiency frontier? Or did you weight for some other reason? In lines 441-446 you say “Prioritisations were also run for incremental ecosystem services weights (i.e., 10, 100, 1000) to favour the selection of planning units that provide benefits for the protection of ecosystem services over planning units that provide benefits for biodiversity representation. Higher weights favour the representation of areas that provide a large amount of ecosystem services over areas that are important for biodiversity.” Shouldn’t your ecosystem service data already quantify the relative benefits of ecosystem services spatially? Why are you including additional weighting and if so, how do you decide the weights?

3) Limitations of ecosystem service data and 400 km² grid cell size for capturing relationships between mangroves and benefit for people

I was very excited to read this paper because as I said, it addresses an important issue of targeting conservation investments for biodiversity and societal benefits. However, I was disappointed to see the large grid size (400 km²) over which the modeling was conducted and the lack of original analysis quantifying ecosystem services.

Happily, there are now papers in the literature that assess ecosystem services provided by mangroves at a global scale (e.g., Mendez et al). Another example is Chaplin-Kramer et al. 2019 Science. However, applying and summarizing these existing data layers for ecosystem service assessments at a 400 km² resolution is problematic. It ignores the scale over which social-ecological processes occur and how management actions such as conservation (e.g., prohibition of fishing, harvesting mangroves for fuel, clearing mangroves for development) may influence these processes. For example, Mendez takes into account the spatial relationship between people, property, and mangroves at a much finer scale; however, that relationship will be lost at the 400 km² scale. I don’t have enough information about the fisheries layer to understand how spatial relationships between supply and delivery/demand are accounted for in the analysis, but again, mangroves that provide nursery or adult habitat are not necessarily located in the same location where adult fish are fished. Moreover, conservation actions taken towards mangroves in one location could influence or redistribute ecosystem service benefits to other locations. I do not believe these dynamics can be accounted for in the optimization with static data layers for the ecosystem services. At a minimum more explanation is needed about the assumptions/limitations in the analysis due to the static assessment of ecosystem services at this large a scale.

Minor comments

- Lines 61-74 missing other important benefits of mangroves - tourism, recreation, cultural aspects, filtration/water quality (cite Barbier et al. 2011 Ecological Monographs). Why didn't you include tourism in the analysis?

- Line 96: What data did they use to know a mangrove area was protected? This is a big component of the analysis and should be at least referenced and at most addressed in an additional sentence in the main ms

- Restoration is an important part of conservation and spatial planning, but is not mentioned once in this paper. Needs to be addressed in discussion.

- I may have missed it, but I don't see any discussion of risk to mangroves or spatial patterns of loss. This is an important component of the value of protecting areas and prioritizing where to invest resources.

- Improve introduction and lay out a roadmap for the paper by including your research questions. I think you asked something like 1) how much ecosystem services provided by mangroves currently within protected areas, 2) without considering existing protected areas where are highest priority locations for citing mangroves to generate combination of three ecosystem services? 3) Where should existing network be expanded to achieve multiple human wellbeing benefits? 4) Something about targets. Laying out these questions will help to highlight utility of this analysis and approach at a management scale and also help reader understand the organization of the paper.

- Lines 137-142 – how are you determining “over-represented” vs. moderately vs. under-represented? Just relative to the other ecosystem services? One could argue having 25% of property value at reduced risk from coastal hazards isn't over-represented. Perhaps we should have more? Needs explanation or rewording.

- Fig. 2f – Latin America and the Caribbean – this area is much smaller than Latin America and the Caribbean. Mesoamerican region and Cuba or Central America and Cuba are probably more appropriate descriptions.

- Line 229 – why wasn't carbon efficiently maximized?

- I am surprised by how little correlation there seems to be between population and property value. While I agree that it is very important to look at both of these metrics – so as to identify places where poorer populations can benefit from coastal risk reduction – often times there is a correlation between these. Places where population density is high are often places with higher property values (e.g., cities). Provide more explanation for results in Fig. 4 a/b.

- Line 130-140 – are you only reporting ecosystem services provided by mangroves within currently protected areas? The word protected is confusing here because “people protected” could mean protected from coastal risk reduction or protected because mangroves are protected or both. Clarify this paragraph.

- Line 390 “mangroves coastal protection” – awkward wording; suggest coastal risk reduction – “coastal protection” gives a connotation that ecosystems can fully protect people and infrastructure and that may not necessarily be the case (just as it isn’t the case for hardened shorelines).

Reviewer #3 (Remarks to the Author):

The paper is well-written, and the process is well-described. I would like to see more detail in the methods (much more), but overall, it is one of the most accessible and understandable papers I have read in this arena. I have no editorial issues, aside from finding a better word than huge to describe a quantity and using an actual scale after the word scale.

For me, they need to address 2, 6, and 7 to proceed.

I think Nature Scientific Reports may be a better venue.

Below are questions and concerns.

1. 74 – Should we, at this point, consider and list some of the adverse effects of mangrove forests as we list their well-known assets?

2. 95 – Where does this coarse number come from, and why? (400 km²). It seems very coarse, and I think this is a flaw. Even with your explanation of the methods, it feels too coarse. It just seems to have

come out of thin air. Indeed current computational systems allow for higher resolutions, and the data synthesis argument (other coarse data), I am not sure, holds up for the following reason. Your mangrove is much higher resolution, your WDPA is much higher resolution, and I think your population source is a poor choice (why not use the 1km² global gridded population). Based on your data and the action unit, I believe between 1km² and 10km² is likely the optimum resolution, not 400km². It is a hand-waving statement to say. We used this spatial resolution to reach a reasonable compromise between the accuracy of other spatial datasets used in our analyses. R is more than capable of running in parallel environments on much (much) larger datasets.

3. 95-97 Why half?

4. 179, that is not a scale (40,000 km²). Do you mean 1:40,000 (I am sure you do not).

5. 246 I think you may have a chicken and egg scenario here based on how protected areas are selected.

6. 274, but some of these areas are protected, just not at the level you decide is important (higher level IUCN categories). These higher levels of protection (think Sundurbans / Everglades / Cayapas-Mataje) have these criteria (think carbon storage) because they were protected earlier. Community protection, through programs such as Socio Manglar, seem entirely overlooked. To your total credit, you touch on this issue later but do not seem to address it. Your paper ignores the local aspects of protection and how these have been shown to offer protection at least equal to that of higher-level IUCN protection. I do not mean to be rude here, but I do think it is a very western approach to conservation that you use. That is why “western” countries dominate your more protected locations as you use IUCN higher-level protected status and ignore the myriad of alternates that often exists in lower-income counties. To be fair, you do realize this (~332).

7. 271- I am not convinced of this from the evidence in your paper. Economic activity exists in these “unprotected areas” (compared to non-protected) as they are not protected. Property and people do not exist in mangroved south Florida (think Everglades), for example (or are very limited). Property and people do exist to be protected in mangroved Guayaquil (for example) because it is not a higher-level protected area. This is a flaw I see in your methodology. You point to this yourself in line~298/~306.

8. 326. I disagree, it is difficult, but I think it can be tackled. Cannot, is a strong word.

9. 336 huge?

Replies to reviewers are in blue text.

Reviewers' comments:

Reviewer #1 (Remarks to the Author):

This work spatially identifies areas of significant ecosystem services provided by mangroves. Similar work has been done identifying the value of coastal wetlands for storm protection.

Robert Costanza, Sharolyn J. Anderson, Paul Sutton, Kenneth Mulder, Obadiah Mulder, Ida Kubiszewski, Xuantong Wang, Xin Liu, Octavio Pérez-Maqueo, M. Luisa Martinez, Diane Jarvis, Greg Dee (2021) The global value of coastal wetlands for storm protection, *Global Environmental Change*, Volume 70, ISSN 0959-3780, <https://doi.org/10.1016/j.gloenvcha.2021.102328>.
(<https://www.sciencedirect.com/science/article/pii/S0959378021001072>)

We thank the reviewer for raising this work. The reviewer is indeed correct that our work identifies areas of significant ecosystem services provided by mangroves. Although Constanza *et al.* (2021) identify the value of coastal wetlands for storm protection, our work is markedly different. This is because: (1) our study identifies priority areas for protecting biodiversity as well as ecosystem service provisioning, whereas Constanza *et al.* (2021) do not consider biodiversity protection; (2) we use data for fisheries benefits; and carbon sequestration and not just coastal protection, which is absent in Constanza *et al.* (2021); (3) our study is not an economic analysis, but merely uses one aspect of economics (the value of the coastal protection of mangroves) along with fisheries and carbon sequestration benefits.

Changed. To address this issue, we have amended the Introduction to highlight the novelty of our work compared to Constanza *et al.* (2021) (See L50-52):

“Although efforts have been made to calculate the value of ecosystem services provided by nature (e.g., Constanza *et al.*¹; Constanza *et al.*²), data on ecosystem services are rarely included in conservation plans.”

This paper is solid. I don't have much to critique. My only suggestion is perhaps a brief 2-5 sentence exploration of the spatial issue of ecosystem service value relative to the proximity of services to built up areas. Some values (e.g. biodiversity) might be more important further from built up areas whereas other values manifest because of their proximity to built up areas (e.g. storm protection services).

Changed. Thanks for the positive feedback.

We added the following to the discussion in the caveats section (L409-412):

“Fourth, including the coastal risk reduction and fisheries benefits provided by mangroves can favour the selection of mangrove areas close to urban areas. However, larger benefits for biodiversity could be provided by protected areas distant from urban areas, considering the multiple pressures for biodiversity related to human activities and human-made infrastructure^{3,4}.”

This paper explores the 'triage' ideas of prioritization of biodiversity and ecosystem service benefits. I am very concerned about sins of omission when conducting this sort of analysis. My take on the Ecosystem Services Valuation approach is to help civilization recognize that nature's contributions to people are so large that we must make much more significant efforts to protect nature. This paper suggests that current prioritization strategies focus on mangroves in Asia and this analysis will broaden our ideas as to what needs to be preserved. I applaud this conclusion; however, the adoption of cost-benefit analyses that mimic the methods of economists are prone to the same potential errors of failing to account for all values. Nonetheless, existing literature on the value of ecosystem services emanating from mangroves already suggests that mangrove ecosystems should have high priority for protection (Costanza *et al.*, 2014), preservation, and stewardship. Arguably, this prioritization should not be based solely on economic arguments but economic arguments do support the conclusion.

Robert Costanza, Rudolf de Groot, Paul Sutton, Sander van der Ploeg, Sharolyn J. Anderson, Ida Kubiszewski, Stephen Farber, R. Kerry Turner (2014) Changes in the global value of ecosystem services, *Global Environmental Change*, Volume 26, Pages 152-158, ISSN 0959-3780
<https://doi.org/10.1016/j.gloenvcha.2014.04.002>.

We agree that conservation decisions should not be based solely on ecosystem services and economic arguments. However, we highlight that our prioritisation considers more ecological arguments than economic ones. In fact, areas in the solution meet conservation targets by protecting all 65 mangrove species included in the analysis (see Fig. 3). In our analysis, we also consider ecosystem services data that are not economic, including the carbon sequestration potential of mangroves, and the number of people employed in fisheries.

Changed. We have tried to highlight these non-economic benefits more and the differences with Costanza *et al.* (2014) in the discussion (L373-378):

“By contrast, we identified areas for the implementation of conservation measures that maximise the protection of ecosystem services provided by mangroves and thus have large human benefits. We have thus identified areas that provide extensive ecosystem services but also meet conservation targets for the protection of species and provide direct services to local populations. This approach complements those that focus on the economic benefits of biodiversity protection rather than producing a spatial plan (e.g., Costanza *et al.*).”

—●— Biodiversity
 —●— Biodiversity & ecosystem services
 —●— Fishing
 —●— People
 —●— Properties
 —●— Carbon

Figure 3 – Conservation targets reached and ecosystem services conserved, in a series of prioritisations based on Selection of priority areas considering all mangrove areas

For incremental area budgets (i.e., increasing the area of mangroves protected): **a, c** Conservation targets reached (%) in a spatial prioritisation optimised for biodiversity alone or for both biodiversity and ecosystem services; and **b, d** the percentage of the total of each ecosystem service protected. Prioritisations **a, b**

Selection of priority areas adding to current strictly protected areas

Selection of priority areas adding to all current protected areas

optimise the selection considering all mangrove areas; **c, d**) add the selection to the current system of strictly-protected areas (13.5% of mangrove area, indicated by a dashed vertical line); and **e, f**) add the selection to the current system of all protected areas (43% of mangrove areas, indicated by a dashed vertical line).

Reviewer #2 (Remarks to the Author):

In their manuscript entitled "Priority areas to protect mangroves and maximise ecosystem services", Dabala and colleagues conduct an optimization analysis of biodiversity and ecosystem service benefits of mangroves. They use the analysis to explore the ecosystem service benefits provided in protected areas currently and to identify priority countries and regions in which to target mangrove protection to achieve both biodiversity and ecosystem service benefits in the future. The paper addresses an important topic that deserves more attention – how to target investments in coastal habitat conservation to achieve ecological and societal benefits. However, I find three major issues with this paper that I think preclude its publication in its current form in Nature Communications. The first two can be addressed with major revisions, but the third is more fundamental to the analysis and without more details on the methods, it's unclear to me yet whether this can be resolved.

Major comments:

1) Further address the value of global prioritization analyses for conservation

a. I commend the authors for acknowledging the recent exchange in the literature between Chaplin-Kramer et al and Wyborn et al on the utility / limitations of global prioritization analyses. However, given this discourse the authors should do more to connect the results of their paper to national/regional actions that could be taken.

b. For example, the Discussion lacks any specific examples and fails to contribute new insights to the literature. Many of the topics are generally addressed in Sala *et al.*, 2021. The discussion of Asia needing to conserve mangroves is less helpful because actions are rarely taken at a continental level. Can you provide examples of spatial planning in other regions moving in this direction? Or discuss seeds of progress in countries in Asia that could be built upon with this information?

c. The authors perpetuate the lack of attention to local or regional examples by only citing global papers in lines 50-60. Missed opportunity to cite examples of spatial planning and ecosystem services informing decisions or being applied at regional scales relevant to management:

- White *et al.*, 2012 Ecosystem service tradeoff analysis reveals the value of marine spatial planning for multiple ocean uses. PNAS
- Lester *et al.*, 2013 Evaluating trade-offs among ecosystem services to inform marine spatial planning. Marine Policy
- Arkema *et al.*, 2015 Embedding ecosystem services into coastal planning leads to better outcomes for nature and people. PNAS
- Many countries and regions around the world are engaged in coastal spatial planning or aiming for conservation targets that consider ecosystem services (e.g., the Caribbean Challenge Initiative; ocean and marine spatial planning in the Seychelles etc.) what about discussing some of these?

Changed. Thank you for making this point about focusing more on the local and regional scale. We think it will highlight the value of our global conservation planning to local implementation. We added information in the discussion with specific local and regional examples comparing them with the results of the paper (L338-359):

"In many countries where mangroves thrive, awareness of the intrinsic value of mangroves is increasing and with it, their protection (e.g., increasing mangrove conservation in Brazil⁵) and restoration (e.g., the Indonesian government promising to restore 600,000 ha of mangroves by 2024⁶). Considering that the restoration of mangroves is expensive and not always effective^{7,8}, the

protection of natural mangroves is a more effective conservation solution in areas important for biodiversity and ecosystem services. Further, restored mangroves provide lower ecosystem services than natural and naturally-regenerated^{7,8}. Multiple countries and regions are currently engaged in coastal conservation planning to expand their protected areas system, and some are trying to include ecosystem services (e.g., the Caribbean Challenge Initiative⁹ and The Seychelles Marine Spatial Plan Initiative¹⁰). Although there are multiple examples in the literature of conservation planning that include ecosystem services at a regional scale^{11–13}, almost all initiatives calculate the values of services as co-benefits and do not include them in the process for the selection of protected areas. Our results show that including ecosystem services in the selection of areas for protection provides substantial benefits and can be considered when implementing new protected areas systems. Countries in the analysis where large areas were selected as high priorities for conservation are obvious places for expansion, building on the encouraging results that have been observed in multiple countries where protection is increasing¹⁴. The identification and protection of the areas providing greater ecosystem services can also be used to incentivise the enforcement of protected areas, highlighting the benefits to people. For example, our results could inform an expansion of protected areas in high-priority areas in Asian countries such as Myanmar, Malaysia and Vietnam, where we found few protected areas and where mangroves are threatened by deforestation for aquaculture and agriculture¹⁵.”

We have also cited local and regional scale examples in the Introduction (L60-62):

“At local and regional scale, the same trend is observed, with multiple studies including ecosystem services in their analysis or evaluating trade-offs between ecosystem services^{11–13}.”

2) Need more information about the methods for calculating each ecosystem service, the optimization analysis, and the targets

a. Targets

- Lines 103-104 – provide more explanation/justification for your approach to developing targets in the intro and methods.

Changed. We provided more explanation/justification on our approach to develop the targets in the introduction (L123-127):

“We assigned conservation targets (*i.e.*, the minimum area to protect for each species) based on the area of the global distribution of mangrove species using a log₁₀-linear interpolation following Rodrigues *et al.*¹⁶ (see Methods). Hence, species with larger distribution ranges were assigned lower targets than those with smaller ranges.”

And in the Methods (L469-480):

“Following Rodrigues *et al.*¹⁶, the conservation target for each species was calculated using a log₁₀-interpolation between a maximum target (100% of the distribution for species with a range <10,000 km²) and a minimum target (10% of the distribution for species with a range >250,000 km²). This choice tries to counterbalance the bias towards the larger representation of more widespread species when using a constant representation target. In fact, species with a smaller distribution range require more-demanding conservation targets, considering they tend to be less abundant and thus more vulnerable to natural events and anthropogenic activities¹⁶. Although this approach of setting targets is somewhat arbitrary, it is common in multiple studies (*e.g.*, Runge *et al.*¹⁷, Hanson *et al.*¹⁸) and is used by international organisations (*e.g.*, Claes *et al.*¹⁹). In local studies, other methods to set the targets, such as the use of fixed area-based policy targets²⁰, minimum viable population analysis²¹ and expert-based thresholds²², could be used.”

- Lines 182-183 – your description of the radar plot seems to use conservation targets to include biodiversity and ecosystem services. Is that correct? If so, ms needs a clearer explanation of what is meant by targets in the paper (from the intro I thought it was just related to biodiversity) and

consistent use of the terminology throughout. Similarly, define what is meant by efficiency in this paper. If you are going to use 944 as a reference point for maximum targets met you need to provide a clear explanation of what that 944 means.

Changed. Yes, that is indeed correct. We now more clearly define what a target is in the Introduction and why there are 944 targets (L119-125):

“To account for the variability in structure, height, and species diversity of mangroves across different climates, tidal amplitudes, and geomorphic settings, we split the species distribution of each of the 65 mangrove species across the biophysical typologies²³ and marine provinces of the world²⁴, resulting in 944 sub-species distributions each assigned a specific conservation target. We assigned conservation targets (*i.e.*, the minimum area to protect for each species) based on the area of the global distribution of mangrove species using a log₁₀-linear interpolation following Rodrigues *et al.*⁴² (see Methods).”

We also defined efficiency the first time we introduce the term in the manuscript (L208-210):

“There was also extensive variability at the continental and country scale in the efficiency (*i.e.*, percentage of targets reached per area of mangroves protected) of current protection of priority areas.”

b. Ecosystem services

- There is nothing in main ms about your approach to quantifying ecosystem services. I was expecting a quantitative modeling approach. In the introduction, include in the overall approach that you are using existing published data layers for the current magnitude of ecosystem services so the reader isn't expecting new modeling/assessment.

Changed. We included in the Introduction that we are using existing published data layers for the current magnitude of ecosystem services (L110-112):

“We generated spatial prioritisations using the *prioritizr* R package²⁵ that maximise the protection of 65 mangrove species (see Methods) and include mangrove ecosystem services from the best available global datasets.”

- Need to give substantially more information about how ecosystem services were estimated for each input dataset into the analysis. The reader shouldn't be expected to read several other papers to understand the information that went into the analyses in this paper and the assumptions for each service.

Changed. We included further information on each layer included in the analysis in the Methods. We described the data from Sanderman *et al.* (2018) and Simard *et al.* (2019) (L527-539):

“We used the estimated aboveground biomass of mangroves from Simard *et al.*²⁶. They calculated the global distribution of aboveground biomass of mangroves, linking field-measured biomass-height allometry with the Shuttle Radar Topography Mission 30 m resolution global digital elevation model of mangrove basal area-weighted height (*i.e.*, the individual tree heights weighted in proportion to their basal area). Soil carbon data were obtained from a machine learning-based model of organic carbon density by Sanderman *et al.*²⁷. Soil carbon was estimated as a function of depth, with an initial estimate of the 0–200 cm organic carbon stock from the global SoilGrids 250m model, and a suite of spatially-explicit covariate layers (*i.e.*, vegetation characteristics, digital elevation data, long-term mean monthly sea surface temperature, tidal elevation amplitude, mean monthly total suspended matter and mangrove typology). The organic carbon density was then used to calculate the mangrove soil carbon calculated for the global extent of mangroves for the year 2000 by summing the organic carbon stock in each pixel.”

We also described the data from Menéndez *et al.* (2020) paper on coastal protection services and their limitations in a better way in the Methods (L509-524):

“The number of people and the value of the properties (USD) safeguarded from flooding by mangroves per 20 km of coastline were obtained from Menéndez *et al.*²⁸. These data were calculated by comparing flood impact scenarios with and without mangroves, combined with the global distribution of people and properties and a corresponding damage function, using the Philippines as a pilot case. We used the benefits of mangroves for coastal risk reduction assessed in terms of total avoided damages to people and property during regular climate events and extreme events. The value for each planning unit was assigned using the nearest neighbour method. Values were then transformed to risk reduction per km² of mangroves by dividing the number of people and the values of property safeguarded from flooding by mangroves, as reported by Menéndez *et al.*²⁸, by the total mangrove area. This area was defined as the total area of mangroves in all planning units that shared the same value of coastal risk reduction after applying the nearest neighbour method (*i.e.*, one planning unit that intersected the data point from Menéndez *et al.*²⁸ plus the closer planning units that were missing a value before applying the nearest neighbour method). For each planning unit, the total number of people and properties safeguarded from flooding was then calculated by multiplying the value per km² of mangroves by the area of mangroves in that planning unit.”

Similarly, we described the data from Zu Ermgassen *et al.* (2021) paper on fishing intensity in mangrove areas and their limitations in the methods (L543-549):

“Fishing intensity from Zu Ermgassen *et al.*²⁹ was calculated as the mean value of the number of fisher days km⁻² year⁻¹ for each planning unit. The authors developed a conceptual model that calculated mangrove fishing intensity using data on the number of mangrove fishers (commercial and non-commercial small-scale inshore fisheries), local non-urban population, accessibility to markets, and catchability as a result of weather. These variables were identified based on an expert-based, anonymous, three-round Delphi method.”

- For example, did the Mendez paper account for differences among mangrove taxa or other habitat characteristics in quantifying the coastal protection service? Need more information about nearest neighbor method. Does your analysis take into account the spatial interconnections between service production and delivery? If so, how?

Changed. We have clarified the Menéndez *et al.* (2020) approach (L415-418).

“Fifth, all ecosystem services datasets (*i.e.*, Sanderman *et al.*²⁷; Simard *et al.*²⁶; Menéndez *et al.*²⁸; Zu Ermgassen *et al.*²⁹) do not consider that different mangrove species and assemblages provide different benefits³⁰ and that cryptic ecological degradation could result in the loss of ecosystem services provisioning^{30,31}.”

Also, spatial interconnections are not taken into account in our analysis. We agree that including it could be relevant for fisheries and carbon sequestration considering that these services can be gained by people living in different places with respect to those where the service is produced. However, in that case, direct benefits can also be gained by the providers of the benefit (*i.e.*, public or private) that sell the product, in this case, the fishing products or the carbon sequestered by mangroves (*e.g.*, through the selling of the fishing product or carbon credits). We better explain this in the manuscript in the methods section (L504-507):

“For all ecosystem services included in the analysis, spatial interconnection between production and delivery of the services was not considered given the local benefits that all these services ensure (*i.e.*, local coastal risk reduction, profits from the sale of seafood and carbon credits).”

We included more information on the nearest neighbour method in the text (L495-504):

“Since some of these datasets were based on different underlying mangrove datasets, we interpolated values from each of the datasets to our planning units using the nearest

neighbourhood method. Specifically, after intersecting the datasets with the planning units, those with missing values were assigned the value of the nearest planning unit with data. Other methods are available (e.g., bilinear interpolation used in Sala *et al.*³²), although they all have their own assumptions. When higher-resolution spatial data on biodiversity distribution and ecosystem services are available, our analyses could be improved. We only included three of the multiple ecosystem services provided by mangroves. Future research, that uses the same method we applied here, could focus on evaluating and including other services, such as tourism, when data become available.”

Also, we better define how we filled missing data from Menéndez *et al.* (2020) (L515-524):

“The value for each planning unit was assigned using the nearest neighbour method. Values were then transformed to risk reduction per km² of mangroves by dividing the number of people and the values of property safeguarded from flooding by mangroves, as reported by Menéndez *et al.*²⁸, by the total mangrove area. This area was defined as the total area of mangroves in all planning units that shared the same value of coastal risk reduction after applying the nearest neighbour method (*i.e.*, one planning unit that intersected the data point from Menéndez *et al.*²⁸ plus the closer planning units that were missing a value before applying the nearest neighbour method). For each planning unit, the total number of people and properties safeguarded from flooding was then calculated by multiplying the value per km² of mangroves by the area of mangroves in that planning unit.”

- Similarly, there are major limitations in using fishing effort data. What fisheries are included/excluded from this data layer? How will fishing effort be redistributed with MPAs? How do different taxa, morphological characteristics taken into account in the biodiversity part of the analysis relate to fisheries benefits in the ecosystem service assessment?

Changed. We agree that there are some limitations in using fishing effort data as proxies for the benefits provided by mangroves. However, we are not targeting the protection of a portion of the sea, but of the mangroves from anthropogenic activities. We specify this by providing additional text in the Introduction (L96-98):

“To evaluate the benefit of mangroves to fisheries, our approach does not consider the closure of ocean areas from fishing but just the protection of mangroves from clearing to preserve their functioning and the increase in catches that result.”

We also included what fisheries are included/excluded from the data layer (see the response to the question requiring more information on the ecosystem services estimation) (L545-549):

“The authors developed a conceptual model that calculated mangrove fishing intensity using data on the number of mangrove fishers (commercial and non-commercial small-scale inshore fisheries), local non-urban population, accessibility to markets, and catchability as a result of weather.”

Different taxa and morphological characteristics were not taken into account in the analysis. We agree that it would be useful to consider how different mangrove species and assemblages provide different benefits for fisheries. However, this was not directly considered in Zu Ermgassen *et al.* (2021). We describe that in the manuscript (L413-416):

“Fifth, all ecosystem services datasets (*i.e.*, Sanderman *et al.*²⁷; Simard *et al.*²⁶; Menéndez *et al.*²⁸; Zu Ermgassen *et al.*²⁹) do not consider that different mangrove species and assemblages provide different benefits³⁰ and that cryptic ecological degradation could result in the loss of ecosystem services provisioning^{30,31}.”

- How does management in the existing MPAs and conservation interventions relate/influence ecosystem services? Do the existing protected areas prohibit impacts to mangroves (e.g.,

harvesting, development)? Do they prohibit fishing? If the MPAs prohibit fishing, how is this handled in the analysis?

Changed. We included a more detailed description of how existing protected areas influence ecosystem services at the beginning of the fourth paragraph of the Introduction (L89-93):

“Protection of mangrove ecosystems has many positive effects, including species conservation, preservation of carbon sinks, and enhanced spillover effects leading to increased catch outside of protected areas^{33,34}. Area-based conservation efforts can protect mangroves from clearing³⁵, maintain the ecosystem services provided, and enhance resilience to climate change^{36,37}.”

We agree that the fact that some MPAs prohibit fishing can be problematic in our analysis considering that we focus on the protection of mangroves. It was not possible to distinguish no-take MPAs that protect or do not mangroves on the land because this information is not reported in the WDPA database, so we did not remove any of those. Also in the data from Zu Ermgassen *et al.* (2021), the protected status was not taken into account when developing the dataset, so the fishing effort in no-take marine protected areas could have been overestimated. We include this in the caveats section (L416-418):

“Sixth, data from Zu Ermgassen *et al.*²⁹ did not consider protection status when estimating fishing intensity in mangrove areas. This could have potentially overestimated the benefits for fishing in areas that are no-take marine protected areas.”

c. Optimization

- Need more explanation of lines 241-245. This is the first time that the ms mentions “weighting.” More explanation may be in the methods, but if you are going to mention weighting in the results/discussion then you have to introduce it earlier in the ms and give the reader some idea of how weighting fits in. What is meant by “In analyses”? Which analyses?

- Did you use the weighting to explore different societal preferences for combinations of services and biodiversity along an efficiency frontier? Or did you weight for some other reason? In lines 441-446 you say “Prioritisations were also run for incremental ecosystem services weights (*i.e.*, 10, 100, 1000) to favour the selection of planning units that provide benefits for the protection of ecosystem services over planning units that provide benefits for biodiversity representation. Higher weights favour the representation of areas that provide a large amount of ecosystem services over areas that are important for biodiversity.” Shouldn’t your ecosystem service data already quantify the relative benefits of ecosystem services spatially? Why are you including additional weighting and if so, how do you decide the weights?

Changed. Thank you for making this point. We agree that we did not sufficiently explain why we included weights in the analysis. We have now included at the end of the introduction (L137-140):

“We also ran weighted prioritisations that favoured the selection of areas that provide more ecosystem services over areas that are important for biodiversity but provide fewer services (see Methods). We then analysed the efficiency of these prioritisations in reaching the conservation targets and providing ecosystem services.”

We clarify this in the Results (L287-290):

“In prioritisations that weight (*i.e.*, increase the probability of selection of a certain feature) the ecosystem services more than biodiversity, we observed an even steeper rise in the services selected at the beginning of the prioritisation (Supplementary Fig. 11).”

We further explain this in the Methods (L577-584):

“To explore uncertainty in our analyses, prioritisations were also run for incremental ecosystem services weights (*i.e.*, 10, 100, 1000) to favour the selection of planning units that provide benefits for the protection of ecosystem services over planning units that provide benefits for biodiversity

representation. Higher weights favour the representation of areas that provide more ecosystem services over areas that are important for biodiversity. Weights used here are indicative and arbitrary, and further research in the effects of weighting ecosystem services could be carried out for analysis at the management level to make the best conservation decisions.”

3) Limitations of ecosystem service data and 400 km² grid cell size for capturing relationships between mangroves and benefit for people

I was very excited to read this paper because as I said, it addresses an important issue of targeting conservation investments for biodiversity and societal benefits. However, I was disappointed to see the large grid size (400 km²) over which the modelling was conducted and the lack of original analysis quantifying ecosystem services.

Happily, there are now papers in the literature that assess ecosystem services provided by mangroves at a global scale (e.g., Mendez et al). Another example is Chaplin-Kramer *et al.*, 2019 Science. However, applying and summarizing these existing data layers for ecosystem service assessments at a 400 km² resolution is problematic. It ignores the scale over which social-ecological processes occur and how management actions such as conservation (e.g., prohibition of fishing, harvesting mangroves for fuel, clearing mangroves for development) may influence these processes. For example, Mendez takes into account the spatial relationship between people, property, and mangroves at a much finer scale; however, that relationship will be lost at the 400 km² scale. I don't have enough information about the fisheries layer to understand how spatial relationships between supply and delivery/demand are accounted for in the analysis, but again, mangroves that provide nursery or adult habitat are not necessarily located in the same location where adult fish are fished. Moreover, conservation actions taken towards mangroves in one location could influence or redistribute ecosystem service benefits to other locations. I do not believe these dynamics can be accounted for in the optimization with static data layers for the ecosystem services. At a minimum more explanation is needed about the assumptions/limitations in the analysis due to the static assessment of ecosystem services at this large a scale.

Changed. We believe there has been a misunderstanding here. In our analysis, we used data on mangrove distribution and ecosystem services with spatial resolution ranging from 25 m to 20 km. We thus defined the final spatial resolution of our grid (~20 km along the coast × 20 km inland to include mangrove areas because they extend inland = 400 km²) based on the coarsest data we included in the analysis (*i.e.*, coastal risk reduction values from Menéndez *et al.* (2020)). We agree that the Menéndez *et al.* (2020) is excellent and have used this dataset in our study.

We did not make this sufficiently explicit in the manuscript and we have now amended text in the Methods. We also provide more explanation about the assumptions and limitations in the analysis due to the static assessment of ecosystem services at this large scale in the Discussion (L439-447):

“Datasets included in this analysis have spatial resolution ranging from 25 m (*i.e.*, global mangrove distribution from Bunting *et al.*³⁸) to 20 km (*i.e.*, coastal risk reduction values from Menéndez *et al.*²⁸). Hence, we standardised all datasets to a (20 km along the coast × 20 km inland to include mangroves that extend inland = 400 km²; Menéndez *et al.*²⁸). Higher resolution data on mangrove species and ecosystem services are not available. For example, Chaplin-Kramer *et al.*³⁹ report data on the number of people safeguarded from flooding by mangroves (and other habitats) at a much coarser 1° resolution (~12,300 km² at the equator). The use of relatively coarse 20-km alongshore planning units in the analysis also assists in the protection of mangrove patches that are large enough to ensure the provision of services⁴⁰.”

Minor comments

- Lines 61-74 missing other important benefits of mangroves - tourism, recreation, cultural aspects, filtration/water quality (cite Barbier *et al.*, 2011 Ecological Monographs). Why didn't you include tourism in the analysis?

Changed. We now cite other benefits of mangroves in the third paragraph of the introduction (L72-80):

"Finally, the high productivity and physical complexity of mangrove ecosystems make them important nursery grounds for many commercial and non-commercial fish and invertebrate species, providing employment and food for millions of people⁴¹. Although in the current analysis we focused on these ecosystem services (*i.e.*, coastal protection, carbon storage, and nursery grounds for fisheries), mangroves provide many other services, including tourism, wood and non-timber forest products, recreation, socio-cultural services and water filtration, and several disservices (negative effects of nature on people), including those related to health, safety and security, leisure and recreation, and materials (see Dahdouh-Guebas *et al.*⁴² for more details)."

The three ecosystem services were chosen because they had available mangrove-specific global databases. We could not find any global database that summarises the benefits of mangroves to the tourism industry. To the best of our knowledge, the best available global dataset that details tourism-centric ecosystem services of mangroves is Spalding and Parrett⁴³, but this study provides a map of mangrove attractions, not of the value of mangroves for tourism. The authors also state that it is not possible to produce global quantitative data from the results of their analysis. However, we believe that this could be an avenue for future research. We stated in the methods the possibility of adding other datasets in the analysis to incorporate the value of all the ecosystem services provided by mangroves (L501-504):

"We only included three of the multiple ecosystem services provided by mangroves. Future research, that uses the same method we applied here, could focus on evaluating and including other services, such as tourism, when data become available."

- Line 96: What data did they use to know a mangrove area was protected? This is a big component of the analysis and should be at least referenced and at most addressed in an additional sentence in the main ms

Changed. We now cite the WDPA database (L113-117):

"Because there is no accepted definition of what constitutes protection, we have used two separate definitions for existing protection were used, based on the IUCN categories in the World Database on Protected Areas (WDPA)⁴⁴: strictly-protected areas based on IUCN categories I-IV; and less strictly-protected areas based on all IUCN categories from I-VI and unclassified areas."

- Restoration is an important part of conservation and spatial planning but is not mentioned once in this paper. Needs to be addressed in discussion.

Changed. We added the following in the second paragraph of the Discussion (L338-344):

"In many countries where mangroves thrive, awareness of the intrinsic value of mangroves is increasing and with it, their protection (*e.g.*, increasing mangrove conservation in Brazil⁵) and restoration (*e.g.*, the Indonesian government promising to restore 600,000 ha of mangroves by 2024⁶). Considering that the restoration of mangroves is expensive and not always effective^{7,8}, the protection of natural mangroves is a more effective conservation solution in areas important for biodiversity and ecosystem services. Further, restored mangroves provide lower ecosystem services than natural and naturally-regenerated^{7,8}."

- I may have missed it, but I don't see any discussion of risk to mangroves or spatial patterns of loss. This is an important component of the value of protecting areas and prioritizing where to invest resources.

Changed. We expanded in the third paragraph of the discussion (L360-365):

“We also found that many priority areas are close to urban centres where there are higher ecosystem services provided by mangroves. In these areas, mangroves are most at risk from clearing, fragmentation, coastal squeeze, and changes in water quality and sedimentation⁴⁵. Although the need for conservation is evident, implementation is not always straightforward. Future research could also include spatial patterns of loss to highlight areas at higher risk and therefore require swift protection.”

- Improve introduction and lay out a roadmap for the paper by including your research questions. I think you asked something like 1) how much ecosystem services provided by mangroves currently within protected areas, 2) without considering existing protected areas where are highest priority locations for citing mangroves to generate combination of three ecosystem services? 3) Where should existing network be expanded to achieve multiple human wellbeing benefits? 4) Something about targets. Laying out these questions will help to highlight utility of this analysis and approach at a management scale and also help reader understand the organization of the paper.

Changed. We added in the introduction (L83-87):

“Here we: (1) assess how well current protected areas represent mangrove biodiversity and ecosystem services; (2) identify areas that, if efficiently protected, could ensure the conservation of mangrove biodiversity and the most ecosystem services, while selecting the smallest area possible; and (3) identify priority areas that could be considered for the expansion of current mangrove protection.”

- Lines 137-142 – how are you determining “over-represented” vs. moderately vs. under-represented? Just relative to the other ecosystem services? One could argue having 25% of property value at reduced risk from coastal hazards isn’t over-represented. Perhaps we should have more? Needs explanation or rewording.

Changed. We have made it clearer that these terms are in relation to the 13.5% of the coverage of the global distribution of mangroves (L172-176).

“A variable percentage of each ecosystem service is conserved in the 13.5% of global mangrove distribution that is currently in protected areas (Fig. 1b-e). Coastal property values are well represented in the current system, which protects 24.9% of the total value globally (amounting to 12.0 billion USD), and stored carbon is moderately-well represented, protecting 14.5% (980.8 Tg C).”

- Fig. 2f – Latin America and the Caribbean – this area is much smaller than Latin America and the Caribbean. Mesoamerican region and Cuba or Central America and Cuba are probably more appropriate descriptions.

Changed to:

“Inset maps (planning units at the analysis resolution of 20-km alongshore) show results for: **b)** Southeast Asia; **c)** The Red Sea; **d)** West Papua, Papua New Guinea, Northern Australia; **e)** Indonesia; and **f)** Mesoamerica and Cuba.”

- Line 229 – why wasn’t carbon efficiently maximized?

Changed. Thank you for pointing this out. We have removed the text between brackets, carbon is actually efficiently maximised, but the lower variation in magnitude compared to the other services results in a near-linear increase as described later in the manuscript (L278-281):

“By contrast, the frequency distribution of carbon sequestration shows that it varies by much less than an order of magnitude, with most values in a tight range (Supplementary Fig. 10c), resulting in

a near-linear increase in the amount of stored carbon conserved as the protected area system is expanded (Fig. 3b, d, f).”

- I am surprised by how little correlation there seems to be between population and property value. While I agree that it is very important to look at both of these metrics – so as to identify places where poorer populations can benefit from coastal risk reduction – often times there is a correlation between these. Places where population density is high are often places with higher property values (e.g., cities). Provide more explanation for results in Fig. 4 a/b.

Changed. We expanded our description of the results reported in Fig. 4 a/b in the last paragraph of the results (L299-306):

“On the other hand, we found much higher mean values of people safeguarded from flooding and fishing intensity in planning units selected by the prioritisation relative to those currently protected (Fig. 4b, d). This is probably because protected areas have been preferentially placed in areas with lower conflicts with fisheries and people, but also be related to the capability of protected areas to reduce fishing pressure and prevent the presence of people (Fig. 4b, d). The weak correlation between the value of properties and the number of people safeguarded from flooding by mangroves (Fig. 4a, b) is likely to be a consequence of the difference in the value of properties between rich and poor countries.”

- Line 130-140 – are you only reporting ecosystem services provided by mangroves within currently protected areas? The word protected is confusing here because “people protected” could mean protected from coastal risk reduction or protected because mangroves are protected or both. Clarify this paragraph.

Changed. We now use safeguarded from flooding throughout the text. We report here an example (L165-179):

“Mangroves inside existing protected areas provide extensive ecosystem services in different countries: greatest values of property currently safeguarded from flooding are in Taiwan (7.8 billion USD) and Mexico (2.9 billion USD; Fig. 1b); the most people safeguarded from flooding are in Vietnam (0.25 million people) and Taiwan (0.17 million people; Fig. 1c); the greatest carbon is stored in Indonesia (324.4 Tg C) and the USA (82.1 Tg C; Fig. 1d); and the most fishers supported by mangroves are in Indonesia (3.4 million fisher days yr⁻¹) and Bangladesh (2.2 million fisher days yr⁻¹; Fig. 1e). A variable percentage of each ecosystem service is conserved in the 13.5% of global mangrove distribution that is currently in protected areas (Fig. 1b-e). Coastal property values are well represented in the current system, which protects 24.9% of the total value globally (amounting to 12.0 billion USD), and stored carbon is moderately-well represented, protecting 14.5% (980.8 Tg C). However, only 6.8% of the total number of people near mangroves are safeguarded from flooding (0.64 million people), and 6.2% of the total fisheries benefit is protected (11.1 million fisher days yr⁻¹). It is clear that the current system of strictly-protected areas was not developed to optimise protection across all ecosystem services but avoided areas where large populations live.”

- Line 390 “mangroves coastal protection” – awkward wording; suggest coastal risk reduction – “coastal protection” gives a connotation that ecosystems can fully protect people and infrastructure and that may not necessarily be the case (just as it isn’t the case for hardened shorelines).

Changed all instances throughout the text to “mangroves coastal risk reduction”.

Reviewer #3 (Remarks to the Author):

The paper is well-written, and the process is well-described. I would like to see more detail in the methods (much more), but overall, it is one of the most accessible and understandable papers I

have read in this arena. I have no editorial issues, aside from finding a better word than huge to describe a quantity and using an actual scale after the word scale.

We thank the Reviewer for the positive comments. They were helpful in strengthening the manuscript. We wanted to highlight that Reviewer #2 also wanted more description of methods and we have expanded our Method section accordingly.

For me, they need to address 2, 6, and 7 to proceed.

I think Nature Scientific Reports may be a better venue.

We believe that this paper is well-suited for publication in Nature Communications as it is a global analysis that provides important advances of significance to specialists in the marine conservation planning field that could lead to a paradigm shift in spatial prioritisation.

Below are questions and concerns.

1. 74 – Should we, at this point, consider and list some of the adverse effects of mangrove forests as we list their well-known assets?

Changed. We also included ecosystem disservices of mangrove forests (L72-82).

"Finally, the high productivity and physical complexity of mangrove ecosystems make them important nursery grounds for many commercial and non-commercial fish and invertebrate species, providing employment and food for millions of people⁴¹. Although in the current analysis we focused on these ecosystem services (*i.e.*, coastal protection, carbon storage, and nursery grounds for fisheries), mangroves provide many other services, including tourism, wood and non-timber forest products, recreation, socio-cultural services and water filtration, and several disservices (negative effects of nature on people), including those related to health, safety and security, leisure and recreation, and materials (see Dahdouh-Guebas *et al.*⁴² for more details). Although mangroves clearly provide substantial benefits to people, priority areas for protection to conserve biodiversity and ecosystem services have not been identified globally."

2. 95 – Where does this coarse number come from, and why? (400 km²). It seems very coarse, and I think this is a flaw. Even with your explanation of the methods, it feels too coarse. It just seems to have come out of thin air. Indeed current computational systems allow for higher resolutions, and the data synthesis argument (other coarse data), I am not sure, holds up for the following reason. Your mangrove is much higher resolution, your WDPA is much higher resolution, and I think your population source is a poor choice (why not use the 1km² global gridded population). Based on your data and the action unit, I believe between 1km² and 10km² is likely the optimum resolution, not 400km². It is a hand-waving statement to say. We used this spatial resolution to reach a reasonable compromise between the accuracy of other spatial datasets used in our analyses. R is more than capable of running in parallel environments on much (much) larger datasets.

Changed. We agree and give more explanation about the assumptions/limitations in the analysis due to the static assessment of ecosystem services at this large scale in the discussion (L442452):

"Datasets included in this analysis have spatial resolution ranging from 25 m (*i.e.*, global mangrove distribution from Bunting *et al.*³⁸) to 20 km (*i.e.*, coastal risk reduction values from Menéndez *et al.*²⁸). Hence, we standardised all datasets to a (20 km along the coast × 20 km inland to include mangroves that extend inland = 400 km²; Menéndez *et al.*²⁸). Higher resolution data on mangrove species and ecosystem services are not available. For example, Chaplin-Kramer *et al.*³⁹ report data on the number of people safeguarded from flooding by mangroves (and other habitats) at a much coarser 1° resolution (~12,300 km² at the equator). The use of relatively coarse 20-km alongshore planning units in the analysis also assists in the protection of mangrove patches that are large enough to ensure the provision of services⁴⁰."

3. 95-97 Why half?

Changed. We have clarified the Discussion (L404-407):

“Third, as boundaries of protected areas do not always line up with the boundaries of our planning units, we assigned a threshold of 50% of the area of mangroves in a planning unit to consider it protected, as has been adopted elsewhere (e.g., Runge *et al.*¹⁷). However, for studies focused on management, it could be defined based on the goal of the prioritisation.”

4. 179, that is not a scale (40,000 km²). Do you mean 1:40,000 (I am sure you do not).

Changed to “resolution” (L217-219):

“Figure 2 – Global mangrove priority areas for protection. a) Priority map, with hexagonal bins aggregated at a resolution of 200-km alongshore for visualisation, shows the median value of the rank of the planning units in the selection (the lower the rank, the more important the planning unit is for protecting biodiversity and delivering ecosystem services).”

5. 246 I think you may have a chicken and egg scenario here based on how protected areas are selected.

Changed. We have now made it clear in the text that we cannot know if the protected areas were defined targeting some services, or if the services provided depend on the presence of the protected area (L293-306).

“We found that mangroves in the current protected areas system provide substantial carbon sequestration and coastal risk reduction for properties, but low amounts of fishing and coastal risk reduction for people (Fig. 4). The mean value of coastal risk reduction for properties, and the mean carbon stored, is similar in planning units currently protected and in those selected by the prioritisation that optimised protection for biodiversity and ecosystem services (Fig. 4a, c). Lower values have been observed when comparing mangroves targeted by less strictly-protected areas compared to those selected by the prioritisation. On the other hand, we found much higher mean values of people safeguarded from flooding and fishing intensity in planning units selected by the prioritisation relative to those currently protected (Fig. 4b, d). This is probably because protected areas have been preferentially placed in areas with lower conflicts with fisheries and people, but could also be related to the capability of protected areas to reduce fishing pressure and prevent the presence of people (Fig. 4b, d). The weak correlation between the value of properties and the number of people safeguarded from flooding by mangroves (Fig. 4a, b) is likely to be a consequence of the difference in the value of properties between rich and poor countries.”

Figure 4 – Mean ecosystem services provided by the existing IUCN I-IV protected areas system (blue), the other existing protected areas (red), and by the proposed expansion to 30% for protection for biodiversity alone (orange) and for biodiversity and ecosystem services (yellow): a) properties and b) population safeguarded from flooding, c) carbon stored, and d) fishing intensity supported by mangroves.

6. 274, but some of these areas are protected, just not at the level you decide is important (higher level IUCN categories). These higher levels of protection (think Sundurbans / Everglades / Cayapas-Mataje) have these criteria (think carbon storage) because they were protected earlier. Community protection, through programs such as Socio Manglar, seem entirely overlooked. To your total credit, you touch on this issue later but do not seem to address it. Your paper ignores the local aspects of protection and how these have been shown to offer protection at least equal to that of higher-level IUCN protection. I do not mean to be rude here, but I do think it is a very western approach to conservation that you use. That is why “western” countries dominate your more protected locations as you use IUCN higher-level protected status and ignore the myriad of alternates that often exists in lower-income counties. To be fair, you do realize this (~332).

Changed. We ran the analysis again while also considering all the protected areas reported in the WDPA database (L113-117):

“Because there is no accepted definition of what constitutes protection, we have used two separate definitions for existing protection were used, based on the IUCN categories in the World Database on Protected Areas (WDPA)⁴⁴: strictly-protected areas based on IUCN categories I-IV; and less strictly-protected areas based on all IUCN categories from I-VI and unclassified areas.”

We included results on the current global distribution of protected areas for mangroves (L144-147):

“The current global system of strictly-protected area covers 13.5% of the distribution of mangroves globally (Supplementary Fig. 1). This increases to 43% of the global distribution of mangroves protected when we also include protected areas under IUCN categories V, VI, and those unclassified (Supplementary Fig. 2).”

Supplementary Fig. 2 – All protected areas (IUCN categories I-VI and unclassified) that intersect global mangrove distribution (red colour).

And how the less strict protection is implemented at national and continental scale (L154-160):

“Similar results were observed at the continental scale when we considered all protected areas, with greatest protection in the Americas (72.4%) compared to other continents (~30%; Supplementary Fig. 3). However, we observed a large variation at the country scale, with 27 countries that protect >70% of their mangroves and 23 that protect <10%. This is the result of many countries having protected areas that are not strictly-protected. For example, there is a large increase in the mangroves considered protected in Brazil, from 12.9% under IUCN categories I-IV to 88.7% under all IUCN protection categories.”

a) Mangrove area (thousands km²)

c) Properties (billions USD)

c) Population (million people)

d) Carbon (Mt)

e) Fishing intensity (million fisher days yr⁻¹)

Supplementary Fig. 3 – Cumulative mangrove area and cumulative ecosystem services, by continent and country, for current protected areas (IUCN I-VI and not classified) and for priority areas added to the current system. a)

Mangrove area and **b-e)** ecosystem services provided by current protected areas (dark shade) and priority areas to expand the current system to 50% coverage (light shade) resulting from a prioritisation that maximises protection of biodiversity and ecosystem services. Only the top-10 countries per continent are shown: **a)** by mangrove area and **b-e)** by amount of ecosystem services provided.

We also included results about the priority areas to protect when expanding the current system of all protected areas (L240-243):

“Similarly, we analysed the expansion of the current system of protected areas (including those that are not strictly-protected) to protect 50% of the global mangrove area (Supplementary Fig. 8). Results are similar, with most priority areas for the expansion in Asia (74.5%) and Oceania (19.5%; Supplementary Fig. 3).”

Supplementary Fig. 8 – Global mangrove priority areas for protection expanding the current protected area system (IUCN I-VI and unclassified). a) Priority map, with hexagonal bins aggregated at a resolution of 200-km alongshore for visualisation, shows the median value of the rank of the planning units in the selection (the lower the rank, the more important the planning unit is for protecting biodiversity and delivering ecosystem services). The radar plot shows the percentage of ecosystem services protected and the percentage of conservation targets reached in the current

protected areas (in black), and in priority areas to expand protection to 50% of the global mangrove area (same colour scale as the map). Inset maps (planning units at the analysis resolution of 20-km alongshore) show results for: **b)** Southeast Asia; **c)** The Red Sea; **d)** West Papua, Papua New Guinea, Northern Australia; **e)** Indonesia; and **f)** Mesoamerica and Cuba.

We expanded Figure 3 adding in e) and f), which detail the results we obtained including other protected areas (L272-273):

“When strictly-protected areas were included in the analysis a larger mangrove area required protection to reach all the conservation targets (Fig. 3e, f).”

—●— Biodiversity
 —●— Biodiversity & ecosystem services
 —●— Fishing
 —●— People
 —●— Properties
 —●— Carbon

Figure 3 – Conservation targets reached and ecosystem services conserved, in a series of prioritisations based on Selection of priority areas considering all mangrove areas
incremental area budgets (i.e., increasing the area of mangroves protected): a, c)

Conservation targets reached (%) in a spatial prioritisation optimised for biodiversity alone or for both biodiversity and ecosystem services; and **b, d)** the percentage of the total of each ecosystem service protected. Prioritisations **a, b)**

Selection of priority areas adding to current strictly protected areas

Selection of priority areas adding to all current protected areas

optimise the selection considering all mangrove areas; **c, d**) add the selection to the current system of strictly-protected areas (13.5% of mangrove area, indicated by a dashed vertical line); and **e, f**) add the selection to the current system of all protected areas (43% of mangrove areas, indicated by a dashed vertical line).

We also included in the Methods why we concentrated mainly on IUCN higher-level protected status in the analysis (L553-558):

“In most analyses, we concentrated on categories I-IV because there is uncertainty in the effectiveness of protected areas having lower IUCN categories because they might be “paper parks” with no effective governance⁴⁶. This is unrelated to the involvement of indigenous people in conservation that have been proven to be valuable⁴⁷⁻⁴⁹. This is also proven by the current conversation in the development of new categories to define different types of protected areas⁵⁰.”

7. 271- I am not convinced of this from the evidence in your paper. Economic activity exists in these “unprotected areas” (compared to non-protected) as they are not protected. Property and people do not exist in mangroved south Florida (think Everglades), for example (or are very limited). Property and people do exist to be protected in mangroved Guayaquil (for example) because it is not a higher-level protected area. This is a flaw I see in your methodology. You point to this yourself in line~298/~306.

Changed. We amend this at the end of the Results (L301-304):

“This is probably because protected areas have been preferentially placed in areas with lower conflicts with fisheries and people, but could also be related to the capability of protected areas to reduce fishing pressure and prevent the presence of people (Fig. 4b, d).”

and we have modified the sentence (L323-325):

“The current protected area system conserves 13.5% of mangroves globally, but is not representative of all mangrove species, and does relatively poorly in protecting high priority conservation areas in our analysis”

8. 326. I disagree, it is difficult, but I think it can be tackled. Cannot, is a strong word.

Changed to:

“Considering that global prioritisation can hardly adequately represent fine-scale variation in the data or local actions⁵¹, the development of systematic conservation plans at local scales are necessary for the effective implementation and management of protected areas⁵².”

9. 336 huge?

Changed to “substantial”:

“Our results show that including ecosystem services in the selection of areas for protection provides substantial benefits and can be considered when implementing new protected areas systems.”

References

1. Costanza, R. *et al.* Changes in the global value of ecosystem services. *Glob. Environ. Change* **26**, 152–158 (2014).
2. Costanza, R. *et al.* The global value of coastal wetlands for storm protection. *Glob. Environ. Change* **70**, 102328 (2021).
3. Mcdonald, R. I. *et al.* Urban effects, distance, and protected areas in an urbanizing world. *Landsc. Urban Plan.* **93**, 63–75 (2009).
4. Tilman, D. *et al.* Future threats to biodiversity and pathways to their prevention. *Nature* **546**, 73–81 (2017).
5. Maretti, C. C. *et al.* Marine and coastal protected and conserved areas strategy in Brazil: Context, lessons, challenges, finance, participation, new management models, and first results. *Aquat. Conserv. Mar. Freshw. Ecosyst.* **29**, 44–70 (2019).
6. Sasmito, S. D. *et al.* Challenges and opportunities for achieving Sustainable Development Goals through restoration of Indonesia’s mangroves. *Nat. Ecol. Evol.* **7**, 62–70 (2023).
7. Schmitt, K. & Duke, N. C. Mangrove Management, Assessment and Monitoring. in *Tropical Forestry Handbook* (eds. Köhl, M. & Pancel, L.) 1–29 (Springer Berlin Heidelberg, 2015). doi:10.1007/978-3-642-41554-8_126-1.
8. Su, J., Friess, D. A. & Gasparatos, A. A meta-analysis of the ecological and economic outcomes of mangrove restoration. *Nat. Commun.* **12**, 5050 (2021).
9. Caribbean Challenge Initiative. <https://caribbeanchallengeinitiative.com> (2023).
10. The Nature Conservancy. *Evaluation of Ecosystem Goods and Services for Seychelles’ Existing and Proposed Protected Area System. An unpublished report to Government of Seychelles – MACCE and SWIOFish3 programme.* 78 (2022).

11. White, C., Halpern, B. S. & Kappel, C. V. Ecosystem service tradeoff analysis reveals the value of marine spatial planning for multiple ocean uses. *Proc. Natl. Acad. Sci.* **109**, 4696–4701 (2012).
12. Lester, S. E. *et al.* Evaluating tradeoffs among ecosystem services to inform marine spatial planning. *Mar. Policy* **38**, 80–89 (2013).
13. Arkema, K. K. *et al.* Embedding ecosystem services in coastal planning leads to better outcomes for people and nature. *Proc. Natl. Acad. Sci.* **112**, 7390–7395 (2015).
14. *The State of the World's Mangroves 2022*. https://www.mangrovealliance.org/wp-content/uploads/2022/09/The-State-of-the-Worlds-Mangroves-Report_2022.pdf (2022).
15. Goldberg, L., Lagomasino, D., Thomas, N. & Fatoyinbo, T. Global declines in human-driven mangrove loss. *Glob. Change Biol.* **26**, 5844–5855 (2020).
16. Rodrigues, A. S. L. *et al.* Global Gap Analysis: Priority Regions for Expanding the Global Protected-Area Network. *BioScience* **54**, 1092 (2004).
17. Runge, C. A. *et al.* Protected areas and global conservation of migratory birds. *Science* **350**, 1255–1258 (2015).
18. Hanson, J. O. *et al.* Global conservation of species' niches. *Nature* **580**, 232–234 (2020).
19. Claes, J. *et al.* *Valuing nature conservation*. (McKinsey & Company, 2022).
20. Friedrichs, M., Hermoso, V., Bremerich, V. & Langhans, S. D. Evaluation of habitat protection under the European Natura 2000 conservation network – The example for Germany. *PLOS ONE* **13**, e0208264 (2018).
21. Taylor, C., Cadenhead, N., Lindenmayer, D. B. & Wintle, B. A. Improving the Design of a Conservation Reserve for a Critically Endangered Species. *PLOS ONE* **12**, e0169629 (2017).
22. Jung, M. *et al.* Areas of global importance for conserving terrestrial biodiversity, carbon and water. *Nat. Ecol. Evol.* **5**, 1499–1509 (2021).

23. Worthington, T. A. *et al.* A global biophysical typology of mangroves and its relevance for ecosystem structure and deforestation. *Sci. Rep.* **10**, 14652 (2020).
24. Spalding, M. D. *et al.* Marine Ecoregions of the World: A Bioregionalization of Coastal and Shelf Areas. *BioScience* **57**, 573–583 (2007).
25. Hanson, J. O. *et al.* *prioritizr: Systematic Conservation Prioritization in R.* (2021).
26. Simard, M. *et al.* Mangrove canopy height globally related to precipitation, temperature and cyclone frequency. *Nat. Geosci.* **12**, 40–45 (2019).
27. Sanderman, J. *et al.* A global map of mangrove forest soil carbon at 30 m spatial resolution. *Environ. Res. Lett.* **13**, 055002 (2018).
28. Menéndez, P., Losada, I. J., Torres-Ortega, S., Narayan, S. & Beck, M. W. The Global Flood Protection Benefits of Mangroves. *Sci. Rep.* **10**, 4404 (2020).
29. zu Ermgassen, P. S. E. *et al.* Reprint of : Fishers who rely on mangroves: Modelling and mapping the global intensity of mangrove-associated fisheries. *Estuar. Coast. Shelf Sci.* **248**, 107159 (2021).
30. Ewel, K., Twilley, R. & Ong, J. Different kinds of mangrove forests provide different goods and services. *Glob. Ecol. Biogeogr. Lett.* **7**, 83–94 (1998).
31. Koedam, N. & Dahdouh-Guebas, F. Ecological quality changes precede changes in quantity in mangrove forests. *Science* (2008).
32. Sala, E. *et al.* Protecting the global ocean for biodiversity, food and climate. *Nature* **592**, 397–402 (2021).
33. Böhnke-Henrichs, A., Baulcomb, C., Koss, R., Hussain, S. S. & de Groot, R. S. Typology and indicators of ecosystem services for marine spatial planning and management. *J. Environ. Manage.* **130**, 135–145 (2013).

34. Di Lorenzo, M., Guidetti, P., Di Franco, A., Calò, A. & Claudet, J. Assessing spillover from marine protected areas and its drivers: A meta-analytical approach. *Fish Fish.* **21**, 906–915 (2020).
35. López-Angarita, J., Tilley, A., Hawkins, J. P., Pedraza, C. & Roberts, C. M. Land use patterns and influences of protected areas on mangroves of the eastern tropical Pacific. *Biol. Conserv.* **227**, 82–91 (2018).
36. Roberts, C. M. *et al.* Marine reserves can mitigate and promote adaptation to climate change. *Proc. Natl. Acad. Sci.* **114**, 6167–6175 (2017).
37. Marcos, C. *et al.* Reviewing the Ecosystem Services, Societal Goods, and Benefits of Marine Protected Areas. *Front. Mar. Sci.* **8**, 613819 (2021).
38. Bunting, P. *et al.* The Global Mangrove Watch—A New 2010 Global Baseline of Mangrove Extent. *Remote Sens.* **10**, 1669 (2018).
39. Chaplin-Kramer, R. *et al.* Global modeling of nature’s contributions to people. *Science* **366**, 255–258 (2019).
40. Kukkala, A. S. & Moilanen, A. Ecosystem services and connectivity in spatial conservation prioritization. *Landsc. Ecol.* **32**, 5–14 (2017).
41. Hutchison, J., Spalding, M. & zu Ermgassen, P. The role of mangroves in fisheries enhancement. *Nat. Conserv. Wetl. Int.* **54**, 434 (2014).
42. Dahdouh-Guebas, F. *et al.* Reconciling nature, people and policy in the mangrove social-ecological system through the adaptive cycle heuristic. *Estuar. Coast. Shelf Sci.* **248**, 106942 (2021).
43. Spalding, M. & Parrett, C. L. Global patterns in mangrove recreation and tourism. *Mar. Policy* **110**, 103540 (2019).
44. UNEP-WCMC & IUCN. *Protected Planet: The World Database on Protected Areas (WDPA)*. (UNEP-WCMC and IUCN, 2021).

45. Turschwell, M. P. *et al.* Multi-scale estimation of the effects of pressures and drivers on mangrove forest loss globally. *Biol. Conserv.* **247**, 108637 (2020).
46. Shafer, C. L. Cautionary thoughts on IUCN protected area management categories V–VI. *Glob. Ecol. Conserv.* **3**, 331–348 (2015).
47. Lele, S., Wilshusen, P., Brockington, D., Seidler, R. & Bawa, K. Beyond exclusion: alternative approaches to biodiversity conservation in the developing tropics. *Curr. Opin. Environ. Sustain.* **2**, 94–100 (2010).
48. Giakoumi, S. *et al.* Revisiting “Success” and “Failure” of Marine Protected Areas: A Conservation Scientist Perspective. *Front. Mar. Sci.* **5**, 223 (2018).
49. Tran, T. C., Ban, N. C. & Bhattacharyya, J. A review of successes, challenges, and lessons from Indigenous protected and conserved areas. *Biol. Conserv.* **241**, 108271 (2020).
50. Grorud-Colvert, K. *et al.* The MPA Guide: A framework to achieve global goals for the ocean. *Science* **373**, eabf0861 (2021).
51. Di Marco, M., Watson, J. E. M., Possingham, H. P. & Venter, O. Limitations and trade-offs in the use of species distribution maps for protected area planning. *J. Appl. Ecol.* **54**, 402–411 (2017).
52. McIntosh, E. J., Pressey, R. L., Lloyd, S., Smith, R. J. & Grenyer, R. The Impact of Systematic Conservation Planning. *Annu. Rev. Environ. Resour.* **42**, 677–697 (2017).

REVIEWER COMMENTS

Reviewer #2 (Remarks to the Author):

The authors have considered the comments from the reviewers and improved the manuscript substantially. However, I do still have several remaining comments.

1. I still think it needs to be made clearer from the beginning of the paper that the prioritization is based on existing ecosystem service datasets, so the reader is not expecting the paper to quantify the marginal change in ecosystem services as a result of implementing MPAs.

Line 110 – reword to...“We generated spatial prioritisations based on mangrove ecosystem service data from the best available global datasets (cite Menéndez et al, zu Ermgassen et al, and carbon study here). We used the prioritizr R package that maximises the protection of 65 mangrove species (see Methods).”

Similarly, in Supplementary Fig. 6, these data are from existing global studies of ecosystem services, right (just aggregated at the 200-km scale)? Thus, I think the legend should reference these data. Say something like:

Supplementary Fig. 6 – Distribution of ecosystem services. Global distribution of: a) value of properties and b) number of people protected by mangroves (from Menéndez et al...); c) carbon stored by mangroves (from X study); and d) fishing intensity in mangroves (from zu Ermgassen et al). Data are aggregated at a 200-km alongshore resolution for visualisation.

This clarification is important for several reasons – 1) setting readers expectations, 2) being clear about the methods, 3) because benefits are not necessarily delivered in the same location where they are produced and thus without modeling we cannot estimate the delivery as a result of protection, 4) because it does reduce the novelty of the paper.

2. The authors have improved the clarity on their methods based on feedback from the reviewers (e.g., on what the targets mean and how they were set). However, there are still a few places that the methods need to be improved.

- Around line 127 – I don't think you ever say anything about how you calculated the summaries of ecosystem services in the existing protected areas (Results reported in lines 165-179). If I'm not mistaken, the authors summed the ecosystem services reported in the previous studies for the locations

within the protected areas? I think it should be made clearer (just need 1-2 sentences) how you estimated these values and again that the underlying values come from previous studies.

- Similarly, the revised ms does provide further details about the previous studies, but in the same way that papers which model ecosystem services provide a summary of each ecosystem service modeled (e.g., see Sala et al 2021 and the new paper I suggested you incorporate below), I think this study should provide more detail in its summaries about how the ecosystem services were quantified in those previous studies, including the limitations of the ES analyses.

- For example, the Zu Ermgassen study estimates fishing intensity. That is a measure of the demand for the service. What is the limitation of using these data as opposed to catch data which combines both supply and demand of the service? Also, I'm unclear from this study about how the Zu Ermgassen study developed a conceptual model that calculates intensity and used it to quantify the service. It is important to provide a little more explanation of the three round Delphi approach and how that works for a global assessment. What data go into this? Similarly, the Menéndez et al paper uses hydrodynamic modeling combined with economic analysis. You need at least a few more sentences for each of these services to explain the modeling approach, data etc. As written, these summaries reflect a lack of understanding of the approach taken in the previous studies (whether this is true or not) and thus not enough information for the reader to understand what was done to produce the underlying information that is at the core of this study.

3. Talk a bit more about why the biodiversity and biodiversity + ES targets track each other in Fig. 3 a, c, e. You report that they do, but don't really explain why. You also say previous studies show a strong relationship with biodiversity and ES. However, this isn't necessarily true and really depends on the service. For example, coastal protection may be greater with multiple lines of nature-based defense (e.g., Guannel et al. 2016), but that is at the taxa, not species level. Furthermore, coastal protection depends more on the height, density, and width of the vegetation or reef and less on the species composition. My main point is that it's actually interesting (and to me somewhat surprising) that these track each other so closely but also begs more discussion of potential mechanisms to be clear that there isn't just a confounding factor here.

4. Is it possible that the reason that the fisheries service doesn't align w/carbon (which would presumably reflect more mangrove habitat for the fishery and thus greater benefits) is because the metric is fishing intensity rather than a metric of production?

Similarly, Line 409 – I disagree that fisheries benefits are greater for MPAs that are closer to urban areas in general....again, this really depends on the data for the ecosystem service and the "serviceshed". In the case of this paper, it is probably true because of the use of fishing intensity rather than catch or another metric that combines both supply and demand.

5. Minor changes:

I suggest citing this new paper at line 66, line 479, line 346 and elsewhere.

Arkema et al. 2023 Evidence-based target setting informs blue carbon strategies for nationally determined contributions. Nature Ecology & Evolution <https://www.nature.com/articles/s41559-023-02081-1>

It deals with quantifying and optimizing mangrove ecosystem services to set (and implement) blue carbon targets (conservation, restoration) for nationally determined contributions. The paper also quantifies potential change in ES as a result of the targets for protection and restoration. Thus, it is an example of leveraging spatial prioritization analyses in collaboration with decision-makers to inform national scale climate goals.

The paper also finds the same attenuating relationship between fisheries and coastal protection benefits with increasing targets as you show in Fig. 3 but for tourism as well. And the same linear shape with carbon. Would be worth a discussion point related to the non-linear relationship between these local ES and mangrove area.

Line 114: check wording

Line 139-140: this is the first place you mention efficiency. I recommend simply stating here how you are interpreting efficiency in this paper.

Reviewer #3 (Remarks to the Author):

Change

Provides substantial benefits and can be considered...

to

Provides benefits and can be considered...

I think this sentence is incorrect. Does not the IUCN has scale-independent (vector) species data?

Higher-resolution data on mangrove species and ecosystem services are not available.

That is all I have.

Replies to reviewers are in blue text.

REVIEWER COMMENTS

Reviewer #2 (Remarks to the Author):

The authors have considered the comments from the reviewers and improved the manuscript substantially. However, I do still have several remaining comments.

1. I still think it needs to be made clearer from the beginning of the paper that the prioritization is based on existing ecosystem service datasets, so the reader is not expecting the paper to quantify the marginal change in ecosystem services as a result of implementing MPAs.

Line 110 – reword to....”We generated spatial prioritisations based on mangrove ecosystem service data from the best available global datasets (cite Menéndez et al, zu Ermgassen et al, and carbon study here). We used the prioritizr R package that maximises the protection of 65 mangrove species (see Methods).”

Changed. See L110-112:

“Using the prioritizr R package¹, we generated spatial prioritisations based on mangrove ecosystem service data from the best available global datasets²⁻⁵. The prioritisations maximised the protection of 65 mangrove species (see Methods).”

Similarly, in Supplementary Fig. 6, these data are from existing global studies of ecosystem services, right (just aggregated at the 200-km scale)? Thus, I think the legend should reference these data. Say something like:

Supplementary Fig. 6 – Distribution of ecosystem services. Global distribution of: a) value of properties and b) number of people protected by mangroves from Menéndez et al.⁵; c) carbon stored by mangroves^{3,4}; and d) fishing intensity in mangroves Zu Ermgassen et al.². Data are aggregated at a 200-km alongshore resolution for visualisation.

This clarification is important for several reasons – 1) setting readers expectations, 2) being clear about the methods, 3) because benefits are not necessarily delivered in the same location where they are produced and thus without modeling we cannot estimate the delivery as a result of protection, 4) because it does reduce the novelty of the paper.

Changed (Supplementary Fig. 6 caption):

“Supplementary Fig. 6 – Distribution of ecosystem services. Global distribution of: a) value of properties and b) number of people protected by mangroves from Menéndez *et al.*⁵; c) carbon stored by mangroves from Simard *et al.*³ and Sanderman *et al.*⁴; and d) fishing intensity in mangroves from zu Ermgassen *et al.*². Data are aggregated at a 200-km alongshore resolution for visualisation.”

2. The authors have improved the clarity on their methods based on feedback from the reviewers (e.g., on what the targets mean and how they were set). However, there are still a few places that the methods need to be improved.

- Around line 127 – I don't think you ever say anything about how you calculated the summaries of ecosystem services in the existing protected areas (Results reported in lines 165-179). If I'm not mistaken, the authors summed the ecosystem services reported in the previous studies for the locations within the protected areas? I think it should be made clearer (just need 1-2 sentences) how you estimated these values and again that the underlying values come from previous studies.

Changed. Added in the Introduction (L128-130):

“We calculated ecosystem services provided by mangroves in protected areas as the sum of the values provided by previous studies²⁻⁵.”

- Similarly, the revised ms does provide further details about the previous studies, but in the same way that papers which model ecosystem services provide a summary of each ecosystem service modeled (e.g., see Sala *et al.* 2021 and the new paper I suggested you incorporate below), I think this study should provide more detail in its summaries about how the ecosystem services were quantified in those previous studies, including the limitations of the ES analyses.

- For example, the Zu Ermgassen study estimates fishing intensity. That is a measure of the demand for the service. What is the limitation of using these data as opposed to catch data which combines both supply and demand of the service? Also, I'm unclear from this study about how the Zu Ermgassen study developed a conceptual model that calculates intensity and used it to quantify the service. It is important to provide a little more explanation of the three round Delphi approach and how that works for a global assessment. What data go into this? Similarly, the Menéndez *et al.* paper uses hydrodynamic modeling combined with economic analysis. You need at least a few more sentences for each of these services to explain the modeling approach, data etc. As written, these summaries reflect a lack of understanding of the approach taken in the previous studies (whether this is true or not) and thus not enough information for the reader to understand what was done to produce the underlying information that is at the core of this study.

Changed. We now provide further details on how the ecosystem services were quantified in the previous studies and their limitations. We describe how the data were produced by Simard *et al.*³ and Sanderman *et al.*⁴ (L555-575):

“We used the estimated aboveground biomass of mangroves from Simard *et al.*³. They calculated the global distribution of aboveground mangrove biomass, linking Light Detection and Ranging (Lidar) heights from the ICESat/GLAS Spaceborne Lidar mission with the Shuttle Radar Topography Mission (SRTM) 30-m resolution global Digital Elevation Model (DEM) of mangrove basal area-weighted height. Field data of basal area weighted height were interpolated with the height data provided by SRTM DEM. The results were then used to produce a global basal area weighted canopy height dataset. These data were finally transformed into aboveground biomass measurements using an allometric equation derived from the field data. Soil carbon data were obtained from a machine learning-based model of organic carbon density by Sanderman *et al.*⁴. They first compiled a mangrove soil carbon database from peer-reviewed literature, grey literature and unpublished data. Then, soil carbon was estimated as a function of depth, an initial estimate of the 0–200 cm organic carbon stock from the global SoilGrids 250-m model downsampled to 30 m resolution⁶, and a suite of spatially-explicit covariate layers (i.e., vegetation characteristics, digital elevation data, long-term mean monthly sea surface temperature, tidal elevation amplitude, mean monthly total suspended matter and mangrove typology). The organic carbon density at different depths was then cumulatively summed to calculate organic carbon stock in each pixel where mangroves were identified. Both studies outlined their limitations. Simard *et al.*³ indicated the possible effect of using different mangrove extent maps on their findings, while Sanderman *et al.*⁴ indicated the possibility of errors and inaccuracy in the database used and the limited information reported about plot locations.”

We also now describe how Menéndez *et al.*⁵ modelled the coastal risk reduction provided by mangroves (L524-543):

“The number of people and the value of the properties (USD) safeguarded from flooding by mangroves per 20 km of coastline were obtained from Menéndez *et al.*⁵. These data were calculated by comparing flood impact scenarios with and without mangroves, combined with the global distribution of people and properties and a corresponding damage function, using the Philippines as a pilot case. Offshore and nearshore dynamics generated by tropical cyclones were simulated in the Philippines and then validated using real data. The results were then used to create a predictive model that used oceanographic variables to predict cyclone parameters. Another model was produced to analyse the role of mangroves in nearshore dynamics. First, offshore hydrodynamic conditions were resolved under both regular climate and tropical cyclones conditions. Then, nearshore dynamics were obtained by propagating regular climate using hybrid downscaling. For tropical cyclones, nearshore dynamics were produced through the regression model derived from the Philippines results. The role of mangroves in attenuating floods was included to obtain the results for scenarios with and without mangroves. Finally, flood maps and an assessment of the consequence of the flooding on people and industrial and residential stock were produced. The caveats listed in Menéndez *et al.*⁵ are related to the limitations of bathymetry, topography and mangroves data, and of the models of the two-dimensional propagations of nearshore waves and storm surges. Further, some countries with <100 ha of mangroves or where benefits were too high (>USD 50,000) were removed from the analysis. In our study, we ran the analysis for mangroves in those countries, but the quantitative results on the amount of coastal protection provided should not be considered.”

We describe how zu Ermgassen *et al.*² modelled fishing intensity in mangroves, better explaining the limitations and the three-round Delphi approach (L579-610):

“Mean fishing intensity was used as a proxy for the benefits of mangroves to fisheries. Fishing intensity from zu Ermgassen *et al.*² was calculated as the mean number of fisher days km⁻² year⁻¹ for each planning unit. zu Ermgassen *et al.*² developed a conceptual model that calculated mangrove fishing intensity using data on the number of mangrove fishers (commercial and non-commercial small-scale inshore fisheries), local non-urban population, accessibility to markets, and catchability as a result of the weather. These variables were identified following an expert-based, anonymous, three-round Delphi method. This method was used to identify the variables gathering consensus across multiple experts that were required to score the importance of each factor in determining mangrove fishing intensity and the certainty in their assessment. The first round was based on an open-ended survey where experts were asked to rank the twenty most important factors determining the volume of fish and invertebrate catch. In the second and third rounds, the same experts were asked to score the importance of the factors (1 = not important to 5 = extremely important) about the fishery type they were most familiar with. Factors that had scores of 1 or 2 by >70% of the experts were removed. A total of 16 factors were selected, but only four had suitable data for the development of the map: number of mangrove fishers, local non-urban population, accessibility to markets and catchability as a result of the weather. The number of small-scale fishers, assumed to be proportional to the local non-urban population, was obtained, on a country level, from a literature review. A “market effect” was included by doubling the number of fishermen per capita for all the areas within 3 h from cities with >50,000 people. Then, the resulting fishermen were assigned to areas of shallow shelf, coral reefs and mangroves based on distance and the ratio 10:10:1. To calculate the number of fish days, zu Ermgassen *et al.*² assumed that fishermen are out at sea 60% of the days. A storm index was calculated to correct the results, considering a 0.75 loss per storm day. This study reported some caveats related to the limited subset of the identified factors that were mappable, the lack of experts with experience in West or Central Africa, the inability to include other structured habitats other than the shallow shelf and coral reefs habitats, the application of global datasets for accounting the impact of storms, and the exclusive focus on near-shore fisheries. Further, using fishing intensity could be misleading because it does not consider the supply of the service considering that the amount of catch could differ among different areas for the same number of days the fishermen are at sea. However, we decided to use these data because, to our knowledge, these are the best available quantitative data on fisheries in mangroves. The study could be improved with finer-scale data when they are available.”

3. Talk a bit more about why the biodiversity and biodiversity + ES targets track each other in Fig. 3 a, c, e. You report that they do, but don't really explain why. You also say previous studies show a strong relationship with biodiversity and ES. However, this isn't necessarily true and really depends on the service. For example, coastal protection may be greater with multiple lines of nature-based defense (e.g., Guannel *et al.* 2016), but that is at the taxa, not species level. Furthermore, coastal protection depends more on the height, density, and width of the vegetation or reef and less on the species composition. My main point is that it's actually interesting (and to me somewhat surprising) that these track each other so closely but also begs more discussion of potential mechanisms to be clear that there isn't just a confounding factor here.

Changed. We now discuss potential mechanisms that could explain why the two solutions track each other (L255-271):

“Remarkably, the efficiency of solutions meeting conservation targets when optimising for both biodiversity and ecosystem services was very similar to the efficiency of solutions when optimising for biodiversity alone (Fig. 3a, c). For example, a similar percentage of targets was reached by the expansion to 30% for the protection of only biodiversity (81.0% of the target) and for biodiversity and ecosystem services (80.0%). Similar results were observed across all area budgets. This seems to be related to higher ecosystem services provisioning in areas of higher biodiversity (here estimated by species richness) that drive the selection of mangrove areas that provide more ecosystem services. This could be explained by higher functional diversity in areas of higher species richness, which can guarantee the provisioning of multiple services.

Nonetheless, in the prioritisation that weighed (i.e., increase the probability of selection of a certain feature) the ecosystem services more than biodiversity, we observed an even steeper rise in the services selected at the beginning of the prioritisation (Supplementary Fig. 10). This could also explain why the prioritisation that targets biodiversity only and the one that targets both biodiversity and ecosystem services track each other. For larger weights, the efficiency in the selection of areas of high ecosystem services provisioning increases, but at the expense of the efficiency in protecting mangrove biodiversity (Supplementary Fig. 10).”

We also agree that the term “biodiversity” is misleading in the discussion in the chapter where we discuss the shift towards including ecosystem services in spatial prioritisation. We now changed it to “protection of biodiversity” (L399-404):

“For example, there is strong evidence that the protection of biodiversity enhances ecosystem services such as coastal protection, carbon sequestration, and enrichment of fisheries in many coastal ecosystems such as coral reefs, salt marshes and seagrasses⁷⁻¹⁰. In these cases, the enhanced management of these systems (including through implementation of protected areas) can be mitigated by the benefits provided by the safeguarded ecosystem services.”

4. Is it possible that the reason that the fisheries service doesn't align w/carbon (which would presumably reflect more mangrove habitat for the fishery and thus greater benefits) is because the metric is fishing intensity rather than a metric of production?

Similarly, Line 409 – I disagree that fisheries benefits are greater for MPAs that are closer to urban areas in general....again, this really depends on the data for the ecosystem service and the “serviceshed”. In the case of this paper, it is probably true because of the use of fishing intensity rather than catch or another metric that combines both supply and demand.

Changed. See L419-426:

“Fourth, in our study, including the coastal risk reduction and fisheries benefits provided by mangroves has favoured the selection of mangroves close to urban areas. However, larger

benefits for biodiversity could be provided by protected areas distant from urban areas, considering the multiple pressures for biodiversity related to human activities and infrastructure^{11,12}. We used data from the best available dataset on fishing intensity in mangroves², but we recognise that the results could differ using catch or another metric that combines both supply and demand. In future studies, other metrics of fisheries productivity could be included to evaluate if the fishing intensity and catch overlap.”

5. Minor changes:

I suggest citing this new paper at line 66, line 479, line 346 and elsewhere.
Arkema et al. 2023 Evidence-based target setting informs blue carbon strategies for nationally determined contributions. Nature Ecology & Evolution
<https://www.nature.com/articles/s41559-023-02081-1>

It deals with quantifying and optimizing mangrove ecosystem services to set (and implement) blue carbon targets (conservation, restoration) for nationally determined contributions. The paper also quantifies potential change in ES as a result of the targets for protection and restoration. Thus, it is an example of leveraging spatial prioritization analyses in collaboration with decision-makers to inform national scale climate goals. The paper also finds the same attenuating relationship between fisheries and coastal protection benefits with increasing targets as you show in Fig. 3 but for tourism as well. And the same linear shape with carbon. Would be worth a discussion point related to the nonlinear relationship between these local ES and mangrove area.

Thank you for suggesting this paper. We are now citing it in the suggested lines. We also cite the paper on L351-354:

“Multiple countries and regions are currently engaged in coastal conservation planning to expand their protected areas system, and some are trying to include ecosystem services (e.g., the Caribbean Challenge Initiative¹³, The Seychelles Marine Spatial Plan Initiative¹⁴ and Belize Nationally Determined Contribution¹⁵).”

Also, as suggested, we changed some text in the Discussion. See L359-363:

“The patterns we found are similar to those reported by Arkema et al.¹⁵. Increases in both fisheries benefits and coastal risk reduction become small for high conservation targets, while carbon sequestration benefits continue to grow linearly. This seems to be related to the non-linear relationship between fisheries benefits and coastal risk reduction with mangrove area¹⁵.”

Line 114: check wording

Thank you for finding this mistake. Changed (L113-117):

“Because there is no accepted definition of what constitutes protection, we have used two separate definitions for existing protection based on the IUCN categories in the World Database on Protected Areas (WDPA): strictly-protected areas based on IUCN categories I-IV; and less strictly-protected areas based on all IUCN categories from I-VI and unclassified areas.”

Line 139-140: this is the first place you mention efficiency. I recommend simply stating here how you are interpreting efficiency in this paper.

Changed.

“We then analysed the efficiency (i.e., percentage of targets reached and amount of ecosystem services provided per area of mangroves protected) of these prioritisations.”

Reviewer #3 (Remarks to the Author):

Change

Provides substantial benefits and can be considered...

to

Provides benefits and can be considered...

Thank you. Changed as suggested.

I think this sentence is incorrect. Does not the IUCN has scale-independent (vector) species data?

Higher-resolution data on mangrove species and ecosystem services are not available.

Changed. We have made it clearer that we are using scale-independent vector species data from the IUCN. However, those vector data report only the broad range of distribution of the species (L468-469):

“We used a variety of data to describe the spatial distribution of mangrove biodiversity. First, we obtained geographic range data of the 65 most-common mangrove species from the IUCN Red List of Threatened Species¹⁶. These scale-independent vector data report only the broad range of distribution of the species.”

That is all I have.

Thank you for your time and effort. Your comments have significantly improved the manuscript.

REFERENCE

1. Hanson, J. O. *et al.* *prioritizr: Systematic Conservation Prioritization in R.* (2021).
2. zu Ermgassen, P. S. E. *et al.* Reprint of : Fishers who rely on mangroves: Modelling and mapping the global intensity of mangrove-associated fisheries. *Estuar. Coast. Shelf Sci.* **248**, 107159 (2021).
3. Simard, M. *et al.* Mangrove canopy height globally related to precipitation, temperature and cyclone frequency. *Nat. Geosci.* **12**, 40–45 (2019).

4. Sanderman, J. *et al.* A global map of mangrove forest soil carbon at 30 m spatial resolution. *Environ. Res. Lett.* **13**, 055002 (2018).
5. Menéndez, P., Losada, I. J., Torres-Ortega, S., Narayan, S. & Beck, M. W. The Global Flood Protection Benefits of Mangroves. *Sci. Rep.* **10**, 4404 (2020).
6. Hengl, T. *et al.* SoilGrids250m: Global gridded soil information based on machine learning. *PLOS ONE* **12**, e0169748 (2017).
7. Hutchison, J., Spalding, M. & zu Ermgassen, P. The role of mangroves in fisheries enhancement. *Nat. Conserv. Wetl. Int.* **54**, 434 (2014).
8. Spalding, M. D. *et al.* The role of ecosystems in coastal protection: Adapting to climate change and coastal hazards. *Ocean Coast. Manag.* **90**, 50–57 (2014).
9. Jänes, H. *et al.* Quantifying fisheries enhancement from coastal vegetated ecosystems. *Ecosyst. Serv.* **43**, 101105 (2020).
10. Jankowska, E., Pelc, R., Alvarez, J., Mehra, M. & Frischmann, C. J. Climate benefits from establishing marine protected areas targeted at blue carbon solutions. *Proc. Natl. Acad. Sci.* **119**, e2121705119 (2022).
11. McDonald, R. I. *et al.* Urban effects, distance, and protected areas in an urbanizing world. *Landsc. Urban Plan.* **93**, 63–75 (2009).
12. Tilman, D. *et al.* Future threats to biodiversity and pathways to their prevention. *Nature* **546**, 73–81 (2017).
13. Caribbean Challenge Initiative. <https://caribbeanchallengeinitiative.com> (2023).
14. The Nature Conservancy. *Evaluation of Ecosystem Goods and Services for Seychelles' Existing and Proposed Protected Area System. An unpublished report to Government of Seychelles – MACCE and SWIOFish3 programme.* 78 (2022).
15. Arkema, K. K. *et al.* Evidence-based target setting informs blue carbon strategies for nationally determined contributions. *Nat. Ecol. Evol.* 1–15 (2023) doi:10.1038/s41559-023-02081-1.
16. IUCN. The IUCN Red List of Threatened Species. Version 2021-3. (2021).